# COVID-Nets: deep CNN architectures for detecting COVID-19 using chest CT scans

Hammam Alshazly[1,2], Christoph Linse[1], Mohamed Abdalla[3,4], Erhardt Barth[1] and Thomas Martinetz[1]

[1] Institut für Neuro- und Bioinformatik, University of Lübeck, Lübeck, Germany
[2] Faculty of Computers and Information, South Valley University, Qena, Egypt
[3] Mathematics Department, Faculty of Science, King Khalid University, Abha, Saudi Arabia
[4] Mathematics Department, Faculty of Science, South Valley University, Qena, Egypt

## ABSTRACT

In this paper we propose two novel deep convolutional network architectures, CovidResNet and CovidDenseNet, to diagnose COVID-19 based on CT images. The models enable transfer learning between different architectures, which might significantly boost the diagnostic performance. Whereas novel architectures usually suffer from the lack of pretrained weights, our proposed models can be partly initialized with larger baseline models like ResNet50 and DenseNet121, which is attractive because of the abundance of public repositories. The architectures are utilized in a first experimental study on the SARS-CoV-2 CT-scan dataset, which contains 4173 CT images for 210 subjects structured in a subject-wise manner into three different classes. The models differentiate between COVID-19, non-COVID-19 viral pneumonia, and healthy samples. We also investigate their performance under three binary classification scenarios where we distinguish COVID-19 from healthy, COVID-19 from non-COVID-19 viral pneumonia, and non-COVID-19 from healthy, respectively. Our proposed models achieve up to 93.87% accuracy, 99.13% precision, 92.49% sensitivity, 97.73% specificity, 95.70% F1-score, and 96.80% AUC score for binary classification, and up to 83.89% accuracy, 80.36% precision, 82.04% sensitivity, 92.07% specificity, 81.05% F1-score, and 94.20% AUC score for the three-class classification tasks. We also validated our models on the COVID19-CT dataset to differentiate COVID-19 and other non-COVID-19 viral infections, and our CovidDenseNet model achieved the best performance with 81.77% accuracy, 79.05% precision, 84.69% sensitivity, 79.05% specificity, 81.77% F1-score, and 87.50% AUC score. The experimental results reveal the effectiveness of the proposed networks in automated COVID-19 detection where they outperform standard models on the considered datasets while being more efficient.

Corresponding author
Hammam Alshazly,
alshazly@inb.uni-luebeck.de

## INTRODUCTION

Coronavirus disease 2019 (COVID-19), a highly infectious disease that affects primarily the respiratory system, is caused by the severe acute respiratory syndrome coronavirus-2

(SARS-CoV-2). The disease has presented massive public health crises and has been declared by the World Health Organization (WHO) as a global pandemic (*Cucinotta & Vanelli, 2020*). By June 13, 2021, there have been 175,306,598 confirmed COVID-19 cases including 3,792,777 deaths, reported to the WHO, and a total of 2,156,550,767 vaccine doses have been administered (*WHO, 2021*).

The virus is still spreading widely at the time and the number of cases is increasing per day in several countries even with the extensive and accelerated pace of vaccinations. The appearance of worrisome new variants that are much more contagious causes several countries to struggle in reducing the number of new cases and necessitates urgent vaccinations. Epidemiologists define the current stage of the pandemic as a race between vaccinations and newly reported cases, particularly the infections caused by more infectious variants. The main challenges in controlling the pandemic severity are the rapid and wide person-to-person transmission of the virus and the spreading of new viral variants across all countries (*Pachetti et al., 2020*). The standard approach to detect SARS-CoV-2 is performed through a virus-specific real-time reverse transcription polymerase chain reaction (RT-PCR) testing. However, RT-PCR testing has several shortcomings, including a low sensitivity rate in the range of 60%–70%, long turnaround times, variabilities in testing techniques, high expenses, and a limited testing capacity in many countries (*Fang et al., 2020*; *Long et al., 2020*). Therefore, the development of other effective and scalable diagnostic tools with higher sensitivity to COVID-19 are of crucial importance and urgently required.

Recent studies have reported that medical imaging of the lungs can be exploited as a suitable alternative testing method for COVID-19. The most widely used imaging modalities for the lungs are the chest radiography (X-ray) and computed tomography(CT). Beside their wide availability in hospitals worldwide, their usage has improved the diagnostic performance and sensitivity for COVID-19 detection (*Kim, Hong & Yoo, 2020*). Nevertheless, comparing the diagnostic accuracy of X-ray and CT in detecting COVID-19, it has been reported that the sensitivity of X-ray is poor, whereas CT scanning has demonstrated higher sensitivity (*Borakati et al., 2020*). Moreover, CT screening has shown to be more sensitive even than RT-PCR testing while being significantly faster and cheaper (*Fang et al., 2020*; *Ai et al., 2020*). According to a study conducted on 1014 COVID-19 patients (*Ai et al., 2020*), RT-PCR could only detect 601/1014 (59%) patients as positives, while the CT-Scan detected 888/1014 (88%) patients as positives. The initial testing for some patients had negative RT-PCR results, whereas the confirmation was inferred based on their CT findings. Furthermore, chest CT screening has been strongly recommended for patients with specific symptoms compatible with viral infections, and their PCR test results are negative (*Kanne, 2020*).

While it might be easy to differentiate patients with COVID-19 from healthy individuals based on CT, it is very challenging to differentiate COVID-19 from non-COVID-19 viral lung infections such as the community acquired pneumonia (CAP) due to two main reasons. First, COVID-19 and other viral infections share similar common patterns and features (*Xu et al., 2021*). Patients with COVID-19 usually manifest several CT radiological features at different locations and distribution patterns such as ground glass opacities

(GGO), consolidation, bilateral infiltration and crazy paving (*Ye et al., 2020*; *Hani et al., 2020*). Second, the CT images may present appearance differences for patients with COVID-19 across different severity (*Yilmaz et al., 2020*). For these reasons, COVID-19 diagnosis from CTs requires interpretation of the CT images by expert physicians and is a labor-intensive, time-consuming, and often subjective. The CTs are first annotated by a practicing radiologist to report the radiographic findings. Then, the findings are analyzed against specific clinical factors to obtain the final diagnosis. During the current pandemic, checking every CT image is not a feasible option as the frontline physicians are faced with a lack of time and massive workload, which increases the physical burden on the staff and might affect the diagnostic quality and efficiency.

Artificial intelligence (AI) techniques and deep convolutional neural networks (CNNs) have the potential to automate COVID-19 detection in patients and assist in the rapid evaluation of CT scans (*Chowdhury et al., 2020*). The powerful representational capability of the deep CNNs can be exploited to differentiate patients with COVID-19 from healthy subjects or others with non-COVID-19 viral infections. Therefore, our study introduces two deep CNN architectures that operate end-to-end to enable automated detection and effective diagnosis of COVID-19 patients based on CT images. The proposed networks have been tailored and validated to differentiate patients with COVID-19, patients with other viral infections, and healthy individuals from the SARS-CoV-2 CT-scan dataset (*Soares et al., 2020*). We also investigated the networks effectiveness in binary classification with all possible class combinations from the considered dataset. Moreover, we validated our models on the COVID19-CT dataset (*Zhao et al., 2020*), and our models provided promising results outperforming state-of-the-art models.

One common issue when using new, non-canonical network architectures is the lack of models that have been pretrained on large-scale datasets like ImageNet (*Deng et al., 2009*). Using pretrained models in transfer learning approaches is attractive when having only small amounts of data to train the model and training a model from scratch would suffer from poor generalization. This is the case especially for image classification tasks for emerging or rare diseases and CNNs with millions of parameters. Starting to train with pretrained models offers initial filters, which are already adapted for visual recognition and only some modifications have to be made to solve the new task. This promotes the eagerness to benefit from transfer learning for COVID-19 detection. Novel architectures, which might be better suited for a specific task, have to compete with the pretrained canonical architectures (i.e., ResNet or DenseNet) and might show inferior performance because of the lack of pretrained weights. Nevertheless, in order to benefit from transfer learning, we designed CovidResNet and CovidDenseNet with parameter compatibility as a key design feature. The idea is to make some network weights compatible with those of pretrained models, which can be found in public repositories. The inter-usability of the weights is realized by sharing some parts of the standard's architecture and adding appropriate adapter layers at certain key positions. As a result, weights from the well known ResNet50 (*He et al., 2016*) or DenseNet121 (*Huang et al., 2017*) architectures can be used to partly initialize CovidResNet and CovidDenseNet, which leads to a boost in performance.

Since the CT images in the considered datasets have different sizes, scaling them to match a fixed input size will probably distort them. We opted for a different preprocessing procedure and experimentally investigated an approach to preserve the aspect ratios of the CT images. This procedure has proved to be very effective and results in an improved overall performance (*Alshazly et al., 2020*). Extensive experiments and analysis on the diagnostic accuracy using standard evaluation metrics were conducted against five standard CNN models. The experimental results show the superior performance for the proposed models over the standard models and are computationally more efficient.

The main contributions of this work are summarized as follows:

- We propose two novel deep CNN architectures (CovidResNet and CovidDenseNet) for automated COVID-19 detection based on chest CT images. The models enable transfer learning between different architectures and they can be partly initialized with larger pretrained models like ResNet50 and DenseNet121, to boost the diagnostic performance. The models have been tailored and validated for the multi-class and binary classification tasks to differentiate COVID-19 patients from non-COVID-19 viral infections as well as healthy subjects.
- The networks are trained and tested on CT images from two benchmark datasets. First, the SARS-CoV-2 CT-scan dataset, which contains 4173 CT images for 210 subjects distributed into three different classes. To the best of our knowledge, this is the first experimental study to be conducted on the SARS-COV-2 CT-scan dataset with a subject-wise data split. Therefore, our models and the reported results may serve as a baseline to benchmark and compare any future work on this dataset. Second, the COVID19-CT dataset, which shares similar visual characteristics and is available in a subject-wise data split.
- In contrast to most of the developed systems that were trained and tested on CT images where the same individuals appear in the training and test splits, which is definitely not appropriate. We followed a subject-wise splitting approach, where we choose 60% of the subjects for training and 40% for testing.
- We conduct extensive experiments and a comprehensive analysis to evaluate the performance of the proposed models for the multi-class and binary classification tasks using various evaluation metrics including accuracy, precision, sensitivity, specificity, F1-score, confusion matrix, ROC curve, and area under the ROC curve (AUC).
- Our experimental results reveal the validity of the proposed networks to achieve very promising results with average accuracies of 93% and 82% for the binary and multi-class classification tasks, respectively. Our CovidResNet and CovidDenseNet models have shown to be effective in differentiating COVID-19 patients from other non-COVID-19 and healthy individuals. Moreover, constructing an ensemble of the proposed networks boosts the performance of the single models and achieves the best results.

The remainder of the paper is structured as follows. 'Related Work' highlights the related work. Our proposed CovidResNet and CovidDenseNet architectures are described in 'COVID-Nets Architectures'. The datasets, data splitting and preprocessing, performance evaluation metrics, and the training methodology are detailed in 'Methodology'. 'Results

and Discussion' provides the experimental results. Finally, the paper is concluded in 'Conclusion'.

## RELATED WORK

This section explores the extensive work on constructing computer-aided diagnostic (CAD) systems for COVID-19 detection based on AI techniques and more specifically deep convolutional networks. Many effective approaches have been proposed to diagnose COVID-19 using chest radiography images including X-rays and CT scans. We hereafter discuss the most relevant work and highlight their success and achieved results.

A considerable number of CAD systems utilise X-ray images to diagnose COVID-19 (*Abraham & Nair, 2020*; *Ibrahim et al., 2021*; *Brunese et al., 2020*; *Aslan et al., 2020*; *Pham, 2021*). For instance, COVID-Net (*Wang, Lin & Wong, 2020*) is a deep CNN model designed specifically for detecting COVID-19 cases from chest X-ray images. COVID-Net was trained and tested on the CVOIDx dataset with a total of 13,975 X-ray images gathered from five different sources of chest radiography images. The network achieved 91.0% sensitivity rate for COVID-19 cases. DeepCoroNet (*Demir, 2021*) is another deep network approach proposed for automated detection of COVID-19 cases from X-ray images. The experimental analysis was performed on a combined dataset of COVID-19, pneumonia, and healthy X-ray images. The model provided a high success rate for the three-class classification problem exceeding other competitive models. CoVNet-19 (*Kedia, Anjum & Katarya, 2021*) is a stacked ensemble model for detecting COVID-19 patients from X-ray images. The model combined two pretrained deep CNNs (VGGNet (*Simonyan & Zisserman, 2015*) and DenseNet (*Huang et al., 2017*)) for feature extraction, and support vector machines (SVMs) for the final classification. The model achieved accuracy of 98.28% and a sensitivity rate above 95% for the COVID-19 class outperforming any of the single models. Coronavirus recognition network (CVR-Net) (*Hasan et al., 2020*) is a multi-scale CNN-based model proposed to recognize COVID-19 from radiography images including both CT and X-ray images. The model was trained and evaluated for the multi-class and two-class classification tasks. The model achieved promising results with average accuracy ranging from of 82% and 99% for the multi-class and binary classification using X-ray images and 78% for CT images. COVID-ResNet (*Farooq & Hafeez, 2020*) is a deep learning approach to differentiate COVID19 cases from other pneumonia cases based on X-ray images. The model was trained and validated on the COVIDx dataset and achieved an accuracy of 96.23%. In *Toraman, Alakus & Turkoglu (2020)*, an artificial neural network approach based on capsule networks was introduced to detect COVID-19 from X-ray images. The proposed model was investigated to differentiate COVID-19 cases in the two-class (COVID-19 and no-findings) as well as multi-class (COVID-19, Pneumonia, and normal) classification tasks. The model achieved average accuracy of 97.24%, and 84.22% for the two-class, and multi-class tasks, respectively.

Similar AI-based systems have been developed for automatically analyzing CT images for detecting COVID-19 pneumonia (*Wu et al., 2021*; *Xu et al., 2020*; *Wang et al., 2021a*; *Li et al., 2020*; *Hasan et al., 2021*). These systems were constructed through a combination

of segmentation and classification models. In the first stage, the lung region or the lesion region are first segmented from the CT scans using segmentation models such as U-Net (*Ronneberger, Fischer & Brox, 2015*) or V-Net (*Milletari, Navab & Ahmadi, 2016*). While in the second stage deep CNN models such as ResNet (*He et al., 2016*) and Attention ResNet (*Wang et al., 2017*) were adopted to perform the diagnosis of COVID-19. For instance, an AI-based system for diagnosing COVID-19 based on CT scans was proposed in *Jin et al. (2020)*. The system was trained and tested on CT-scan dataset consisting of CT scans of different classes including COVID-19, influenza, non-viral pneumonia, and non-pneumonia subjects. A comprehensive analysis was performed on a test cohort to evaluate the performance of the system in the multi-class and binary classification tasks. The model achieved an AUC score of 97.81% and a sensitivity of 91.51% for the multi-class classification task.

At the same time, new deep CNN architectures were designed and adopted to diagnose COVID-19. *Wang, Liu & Dou (2020)* redesigned the COVID-Net architecture and its learning methodology to be applied to CT images, and to improve the prediction accuracy and computational complexity. Besides, a joint learning approach was proposed to improve the diagnostic performance of COVID-19 cases and to tackle the data heterogeneity in the used CT scan datasets. Experiments on two CT image datasets show the success of the proposed joint learning approach with 90.83% accuracy and 85.89% sensitivity on the largest dataset. COVIDNet-CT (*Gunraj, Wang & Wong, 2020*), is deep CNN tailored specifically for the detection COVID-19 cases from chest CT images. The network was designed with a high architectural diversity and lightweight design patterns to achieve high representational capacity and computational efficiency. Training and testing were conducted on a collected CT image dataset named COVIDx-CT, which had CT images for three different classes, including: COVID-19 pneumonia, non-COVID-19 infections, and normal controls. The network achieved high sensitivity and specificity scores for the COVID-19 class reaching up to 97.3% and 99.9%, respectively.

Covid CT-Net (*Swapnarekha et al., 2021*), is a simple deep CNN developed for differentiating COVID-19 CTs from non-COVID-19 CT images. The network was trained and validated on the SARS-CoV-2 CT-scan dataset, which consists of 2492 CT scans for two class: COVID-19 and non-COVID-19 (*Soares et al., 2020*). The experimental results confessed an improved accuracy, specificity, and sensitivity of 95.78%, 95.56%, and 96%, respectively. An attentional convolutional network(COVID CT-Net) to predict COVID-19 from CT images was proposed in *Yazdani et al. (2020)*. The network represented a combination of stacked residual modules empowered with attention-aware units to perform a more accurate prediction. The model was trained and validated on the SARS-CoV-2 CT-scan dataset (*Soares et al., 2020*) and achieved sensitivity and specificity rates of 85% and 96.2% respectively. *Singh, Kumar & Kaur (2020)* proposed a classification model for COVID-19 patients using chest CT images. The model adopted multi-objective differential evolution-based convolutional neural networks to differentiate positive COVID-19 cases from others. Experimental results showed that the proposed model was able to classify the CT images with an acceptable accuracy rate. Zhang et al. proposed a residual learning diagnosis detection network to differentiate COVID-19 cases from other heterogeneous

CT images (*Zhang et al., 2021*). The network was trained and test on the COVID-CT dataset (*Zhao et al., 2020*), and achieved accuracy, precision, and sensitivity of 91.33%, 91.30%, and 90%, respectively. In *Ouyang et al. (2020)*, an attention network was proposed to diagnose COVID-19 from community acquired pneumonia based on CT images. The network was trained and validated on a large-scale CT image dataset collected from eight hospitals. The network testing was performed on an independent CT data, and achieved an accuracy of 87.5%, a sensitivity of 86.9%, and a specificity of 90.1%.

*Jaiswal et al. (2020)* proposed a deep transfer learning approach using a variant of the DenseNet models. The pretrained 201-layer DenseNet model on the IamgeNet dataset was utilized as a base for feature extraction with three added fully connected layers to perform the classification task. The experiments were conducted on the SARS-CoV-2 CT-scan dataset (*Soares et al., 2020*). The model achieved accuracy score of 96.25% and a sensitivity rate of 96.21%. *Alshazly et al. (2021)* conducted experimental study by adopting 12 pretrained deep CNN models, which were fine-tuned using CT images, to differentiate Patients with COVID-19 and non-COVID-19 subjects. Extensive experiments and analysis were performed on two COVID-19 CT scans datasets. The models were trained using custom-sized inputs for each deep model and achieved state-of-the-art results on the considered datasets. Further, visualization techniques were applied to provide visual explanations and show the ability of the fine-tuned models to accurately localize COVID-19 associated regions. In *Pham (2020)*, a similar comprehensive study with 16 pretrained networks was carried out to detect COVID-19 based on CT images. The obtained results were comparable with those achieved in previous reposts as in *Alshazly et al. (2021)*.

Ensemble learning and deep ensembles were also explored in COVID-19 detection to improve the performance of single models. *Singh, Kumar & Kaur (2021)* proposed an ensemble based on three deep networks including: VGGNet (*Simonyan & Zisserman, 2015*), ResNet (*He et al., 2016*), and DenseNet (*Huang et al., 2017*), which were pretrained on natural images. The networks were considered for extracting features from the CT images, and a set of fully connected layers were added on top to perform the classification task. Experiments were conducted on a dataset with CT scans collected from different sources for patients with COVID-19, other lung diseases, and healthy subjects. The proposed ensemble achieved better performance than using any single model from the ensembled networks. (*Zhou et al. (2020)* proposed an ensemble of three pretrained deep CNN models, namely AlexNet (*Krizhevsky, Sutskever & Hinton, 2012*), GoogleNet (*Szegedy et al., 2015*), and ResNet (*He et al., 2016*) to improve the classification accuracy of COVID-19. Experiments were conducted on a collected CT image dataset organized in three different classes, including: COVID-19, lung tumors, and normal lungs. The obtained results showed an improved classification performance for the ensemble compared to any single individual model. *Attallah, Ragab & Sharkas (2020)* proposed a CAD system for distinguishing COVID-19 and non-COVID-19 cases. The system was trained and tested using CT images, where the CT image features were extracted with four pretrained deep CNN models, and then were fused for training support vector machine classifiers. The authors experimented with different fusion strategies to investigate the impact of feature

fusion on the diagnostic performance. The system achieved accuracy, sensitivity, and specificity scores of 94.7%, 95.6%, and 93.7%, respectively.

The above-mentioned techniques were trained and tested on chest radiography images from of the same subjects, and were proposed for differentiating between COVID-19 and healthy individuals. The obtained results need to be validated on datasets that are structured in a subject-wise level and for differentiating COVID-19 from other non-COVID-19 findings, which represents a very challenging task. The reasons are the potential overlap and high visual similarities between the radiographic findings of COVID-19 and non-COVID-19 viral infections. In our study, we develop and test two deep network architectures, which can be partly initialized from standard pretrained models to boost the diagnostic performance. The models are evaluated to differentiate patients with COVID-19 from other non-COVID-19 viral infections as well as healthy individuals. The networks were developed and tested for the multi-class and binary classification tasks. The obtained results are promising and validate the effectiveness of our models. Extensive experiments were conducted on two challenging CT image dataset, which contain images of varying sizes and visual characteristics. The obtained results indicate the success of our models to achieve the best perfromance on the considerd datasets.

Table 1 summarizes some of the recently published studies on detecting COVID-19 from chest radiographical images using various techniques.

## COVID-NETS ARCHITECTURES

Here we describe our proposed CovidResNet and CovidDenseNet models for the automated COVID-19 detection. Inspired by the outstanding performance of the well-designed ResNet (*He et al., 2016*) and DenseNet (*Huang et al., 2017*) architectures, we build our networks by following similar construction patterns to get the benefits of both architectures.

The novelty of CovidResNet and CovidDenseNet lies in the feature-interusability with their standard counterparts, i.e., ResNet50 and DenseNet121. The idea is to initialize the models with pretrained weights from the standard models and benefit from the advantages of transfer learning without the costly step to pre-train models on large-scale datasets like the ImageNet dataset. Nevertheless, one benefits from the tiny model size, parameter efficiency and speed of our architectures, which are designed to have much less weights in total.

Figure 1 illustrates the approach to build and train CovidResNet and CovidDenseNet and how the feature-interusability is enabled. The diagram shows the architectures as a sequence of stacks. CovidResNet contains a convolutional layer, four residual blocks and a fully connected layer. CovidDenseNet also starts with a convolutional layer, followed by four Denseblocks, which have transition layers, and it ends with a fully connected layer. The red faces in Fig. 1 indicate the missing weights in comparison to the standard architectures. As illustrated in the diagram, CovidResNet and CovidDenseNet share subnetworks of their counterparts. More precisely, the first convolutional layer and the first stack are identical to ResNet50 and DenseNet121 and the weights are frozen during training. The subsequent stacks contain much less blocks than the standard models to decrease the total number

**Table 1** **A summary of recently published studies on COVID-19 detection.**

| Literature | Modality | Dataset | Model | Results |
|---|---|---|---|---|
| *Wang et al. (2021c)* | 3,583 X-ray images | Open image data + Xiangya Hospital | Discrimination-DL | Acc. (93.65%)<br>Sens. (90.92%)<br>Spec. (92.62%)<br>AUC (95.5%) |
| *Bahgat et al. (2021)* | 12,933 X-ray images | Several combined datasets | Optimized DenseNet121 | Acc. (98.47%)<br>Prec. (98.50%)<br>Sens. (98.47%)<br>Spec. (99.50%)<br>F1-score (98.49%)<br>AUC (99.83%) |
| *Öztürk, Özkaya & Barstuğan (2021)* | 495 X-ray images | COVID-19 image data collection (*Cohen et al., 2020*) | Shrunken features + PCA + SVM | Acc.(94.2%)<br>Prec. (96.7%)<br>Sens. (93.3%)<br>Spec. (98.5%)<br>F1-score (93.9%) |
| *Alshazly et al. (2021)* | 2,482 CT images | SARS-CoV-2 CT-scan dataset (*Soares et al., 2020*) | ResNet101 | Acc. (99.4%)<br>Prec. (998.6%)<br>Sens. (99.1%)<br>Spec. (99.6%)<br>F1-score (99.4%) |
| *Alshazly et al. (2021)* | 746 CT images | COVID19-CT dataset (*Zhao et al., 2020*) | DenseNet201 | Acc. (92.9%)<br>Prec. (91.3%)<br>Sens. (93.7%)<br>Spec. (92.2%)<br>F1-score (92.5%) |
| *Barstugan, Ozkaya & Ozturk (2020)* | 150 CT images | COVID-19 database (*Di Radiologia Medica e Interventistica, 2021*) | DWT + SVM | Acc. (97.8%)<br>Prec. (98.4%)<br>Sens. (96.8%)<br>Spec. (98.6%)<br>F1-score (97.6%) |
| *Özkaya, Öztürk & Barstugan (2020)* | 150 CT images | COVID-19 database (*Di Radiologia Medica e Interventistica, 2021*) | Fusion of CNN features | Acc. (95.6%)<br>Prec. (97.7%)<br>Sens. (93.3%)<br>Spec. (97.8%)<br>F1-score (95.5%) |
| *Özkaya et al. (2020)* | 2,482 CT images | SARS-CoV-2 CT-scan dataset (*Soares et al., 2020*) | Conv. Support Vector Machine (CSVM) | Acc.(94.0%)<br>Prec. (92.1%)<br>Sens. (96.0%)<br>Spec. (92.0%)<br>F1-score (94.1%) |
| *Ragab & Attallah (2020)* | 2,482 CT images | SARS-CoV-2 CT-scan dataset (*Soares et al., 2020*) | Fusion of handcrafted and CNN features | Acc.(99%)<br>Prec. (99%)<br>Sens. (99%)<br>F1-score (99%)<br>AUC (100%) |
| *Wang et al. (2021b)* | 1,065 CT images | Private CT images coll. from 3 hospitals | Modified Inception model | Acc. (89.5%)<br>Sens. (87%)<br>Spec. (88%)<br>F1-score (77%)<br>AUC (93%) |

| Literature | Modality | Dataset | Model | Results |
|---|---|---|---|---|
| *Song et al. (2021)* | 1,990 CT images | Private CT images coll. from hospitals | DRE-Net | Acc. (86%)<br>Prec. (79%)<br>Sens. (96%)<br>Spec. (77%)<br>F1-score (87%)<br>AUC (95%) |
| *Ardakani et al. (2020)* | 1,020 CT images | Private data | Xception | Acc. (99.02%)<br>Sens. (98.04%)<br>Spec. (100%)<br>F1-score (77%)<br>AUC (93%) |
| *Shi et al. (2021)* | 1,658 patients with COVID-19<br>1,027 patients with CAP | Private CT images collected from hospitals | Handcrafted features | Acc. (89.4%)<br>Sens. (90.7%)<br>Spec. (87.2%)<br>AUC (95.5%) |

**Notes.**
Sens., sensitivity; Spec., specificity; Prec., precision; Acc., accuracy; AUC, area under the curve.

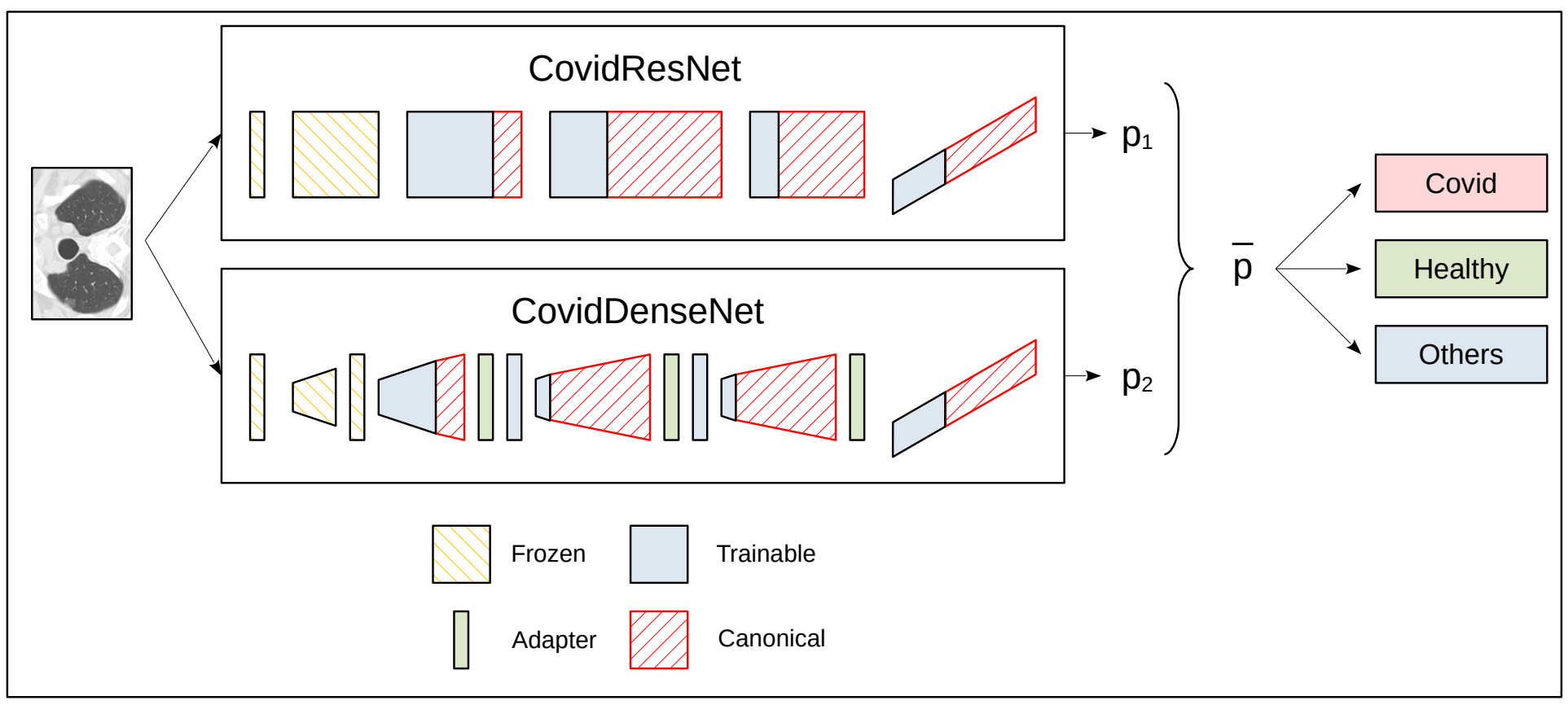

**Figure 1 A schematic diagram for the ensemble prediction process for the three-class problem.** Both networks accept the same input CT image and each network outputs an independent class probability vector. The probability vectors are then averaged to obtain the final predicted class with highest probability.

of features and model complexity. The pruning of the layers leads to a problem for the CovidDenseNet model. A shortening of the dense block is bound to a reduction in the number of channels of the output. This leads to a misfit at the subsequent transition layer because more channels are expected. To solve this problem, we insert an adapter layer after a dense block. The adapter layers consist of a $1 \times 1$ convolution to adjust the number of channels accordingly. More details about our architectures are described in the following subsections.

## CovidResNet

Our CovidResNet architecture is based on the deep residual networks (ResNets) (*He et al., 2016*). ResNet is considered a very deep CNN architecture and the winner of the

2015 ImageNet challenges (*Russakovsky et al., 2015*). The main problems that have been addressed by the ResNet models are the vanishing gradients and performance degradation, which occur during training deep networks. A residual learning framework was proposed, which promotes the layers to learn residual functions with respect to the layer input. While conventional network layers are assumed to learn a desired underlying function $y = f(x)$ by some stacked layers, the residual layers attempt to approximate $y$ via $f(x) + x$. The residual layers start with the input $x$ and evolve to a more complex function as the network learns. This type of residual learning allows training very deep networks and attains an improved performance from the increased depth.

The basic building block for CovidResNet is the bottleneck residual module depicted in Fig. 2. The input signal to the module passes through two branches. The left branch is a stack of three convolutional layers. The first $1 \times 1$ convolution is used for reducing the depth of the feature maps before the costly $3 \times 3$ convolutions, whereas the second $1 \times 1$ is used for increasing the depth to match the input dimensions. The convolutions are followed by batch normalization (BN) (*Ioffe & Szegedy, 2015*) and rectified linear unit(ReLU) (*Nair & Hinton, 2010*) activation. The right branch is a shortcut connection that connects the module's input with the output of the stacked layers, which are summed up before applying a final ReLU activation.

CovidResNet is considered a deep model that consists of 29 layers. The first layer is made of $7 \times 7$ convolutional filters with a stride of 2. Following is a max pooling layer to downsample the spatial dimensions. The architecture continues with a stack of four ResNet blocks, where each block has a number between one and three bottleneck residual modules. When moving from a ResNet block to the next one, the spatial dimension is reduced by max pooling and the number of the learned filters is doubled. In the first block, we stack three modules, each having three convolutional layers with 64, 64 and 256 filters, respectively. After another max pooling layer, we stack three more bottleneck modules with a configuration of 128, 128 and 512 filters, which forms the second block. The same procedure is repeated for the third and fourth blocks, where the former has two stacked modules and the later has only one. The network ends with an adaptive average pooling step and a fully connected layer. Table 2 summarizes the CovidResNet architecture and a visualization is given in Fig. 1. As can be seen in the diagram, the first convolutional layer and the entire first block are frozen during transfer learning. Only the weights of deeper are frozen during transfer learning. Only the weights of deeper layers are adjusted. The diagram also indicates the complimentary layers that exist in the canonical ResNet50 model but not in CovidResNet.

## CovidDenseNet

Our CovidDenseNet model is based on the densely connected network (DenseNet) architectures introduced in *Huang et al. (2017)*. DenseNet addressed the notorious problem of vanishing gradients with a different approach compared to ResNet. Instead of using skip connections to combine the feature maps through summation before passing them to the next layer, the feature maps from all preceding layers are considered as the input to the next layer, and its feature maps are passed to all subsequent layers. The advantages

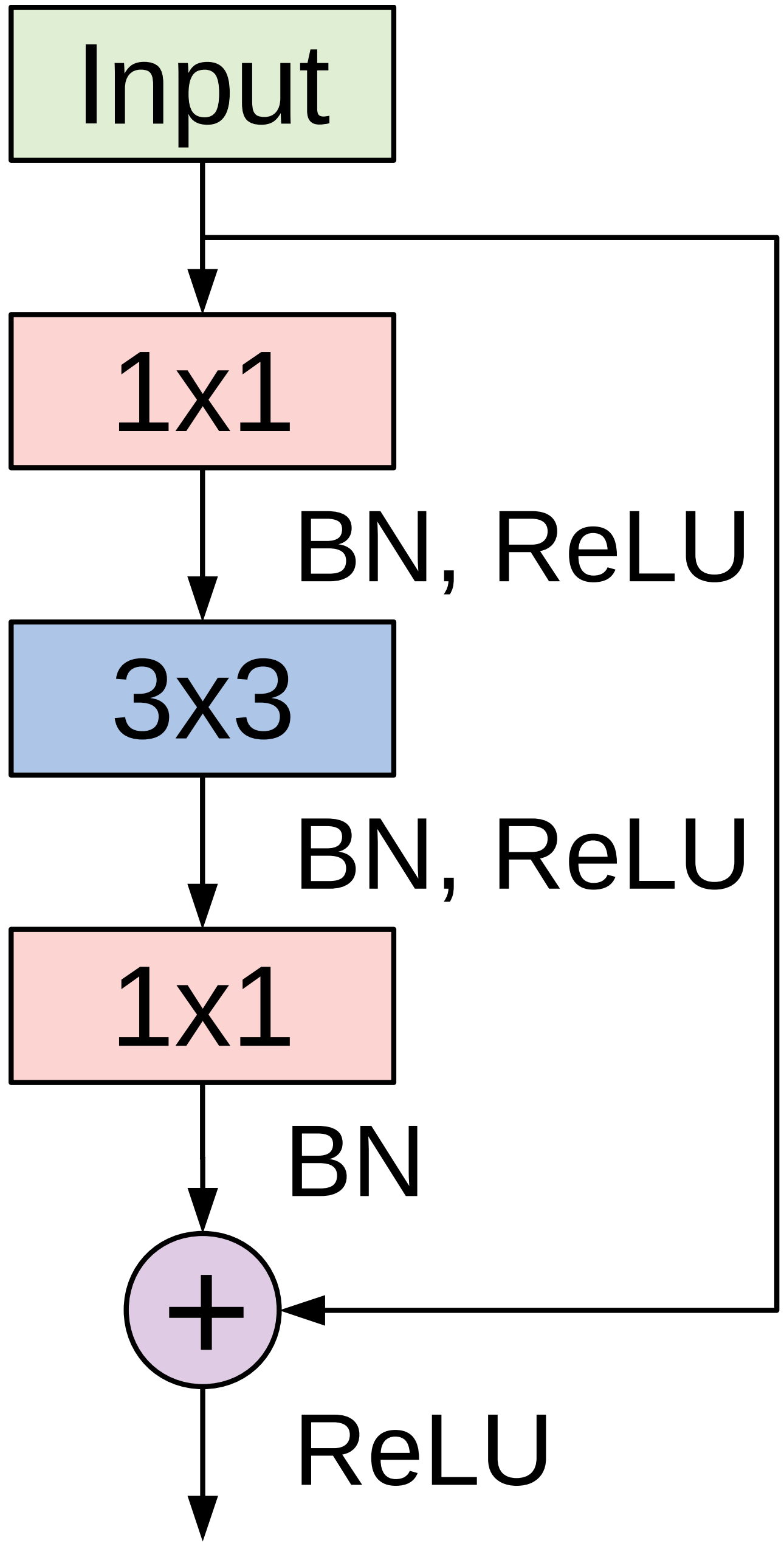

**Figure 2 The bottleneck residual module used in CovidResNet.** The module was first introduced in *He et al. (2016)*.

of the dense connectivity are the improved flow of information throughout the network, where each layer has a direct access to the gradients from the input and the loss function. DenseNets have shown an improved performance for image recognition tasks and are computationally efficient.

The basic building block for the CovidDenseNet model is the DenseNet block. A simplistic form of the dense connectivity of a dense block is shown in Fig. 3. The block has

**Table 2 Description of our CovidResNet architecture for COVID-19 detection.** The network accepts an RGB-input of size 253 × 349 pixels. The residual modules are placed in brackets multiplied by the number of modules stacked per block.

| Layers | Output size | CovidResNet |
|---|---|---|
| Convolution | $129 \times 177$ | $7 \times 7$, 64, stride 2 |
| Pooling | $64 \times 88$ | $3 \times 3$ max pooling, stride 2 |
| ResNet Block (1) | $64 \times 88$ | $\begin{bmatrix} 1 \times 1, 64 \\ 3 \times 3, 64 \\ 1 \times 1, 256 \end{bmatrix} \times 3$ |
| ResNet Block (2) | $32 \times 44$ | $\begin{bmatrix} 1 \times 1, 128 \\ 3 \times 3, 128 \\ 1 \times 1, 512 \end{bmatrix} \times 3$ |
| ResNet Block (3) | $16 \times 22$ | $\begin{bmatrix} 1 \times 1, 256 \\ 3 \times 3, 256 \\ 1 \times 1, 1024 \end{bmatrix} \times 2$ |
| ResNet Block (4) | $8 \times 11$ | $\begin{bmatrix} 1 \times 1, 512 \\ 3 \times 3, 512 \\ 1 \times 1, 2048 \end{bmatrix} \times 1$ |
| Classification layer | $1 \times 1$ | $8 \times 11$ Adaptive average pool fully connected, softmax |

three layers and each layer performs a series of batch normalization, ReLU activation, and $3 \times 3$ convolution operations. The concatenated feature maps from all preceding layers are the input to the subsequent layer. Each layer generates $k$ feature maps, where $k$ is the growth rate. So, if $k_0$ is the input to layer $x_0$, then there are $3k + k_0$ feature maps at the end of the 3-layer dense block. However, two main issues arise as the network depth increases. First, as each layer generates $k$ feature maps, the inputs to layer $l$ will be $(l-1)k + k_0$, and with deep networks this number can grow rapidly and slow down computation. Second, when the network gets deeper, we need to reduce the feature maps size to increase the kernel's receptive field. So, when concatenating feature maps of different sizes we need to match the dimensions. The first issue is addressed by introducing a bottleneck layer of $1 \times 1$ convolution and $4 \times k$ filters after every concatenation. The second issue is addressed by adding a transition layer between the dense blocks. The layer includes batch normalization and $1 \times 1$ convolution followed by an average pooling operation.

To ensure the inter-usability of the weights, CovidDenseNet contains a set of adapter layers consisting of a $1 \times 1$ convolution to increase the number of channels to the size required by the subsequent layer. The number of channels can be seen in Table 3. The last adapter layer is optional. The adapters are inserted between a dense block and the transition layer. Our CovidDenseNet model consists of 43 weighted layers. The first layer is a convolutional layer with $7 \times 7$ filters and uses a stride of 2, followed by a max pooling operation. Then we stack four dense blocks interspersed by transition layers. After the last dense block we perform an adaptive average pooling and add a fully connected layer with a softmax classifier. All details about the CovidDenseNet architecture including the number

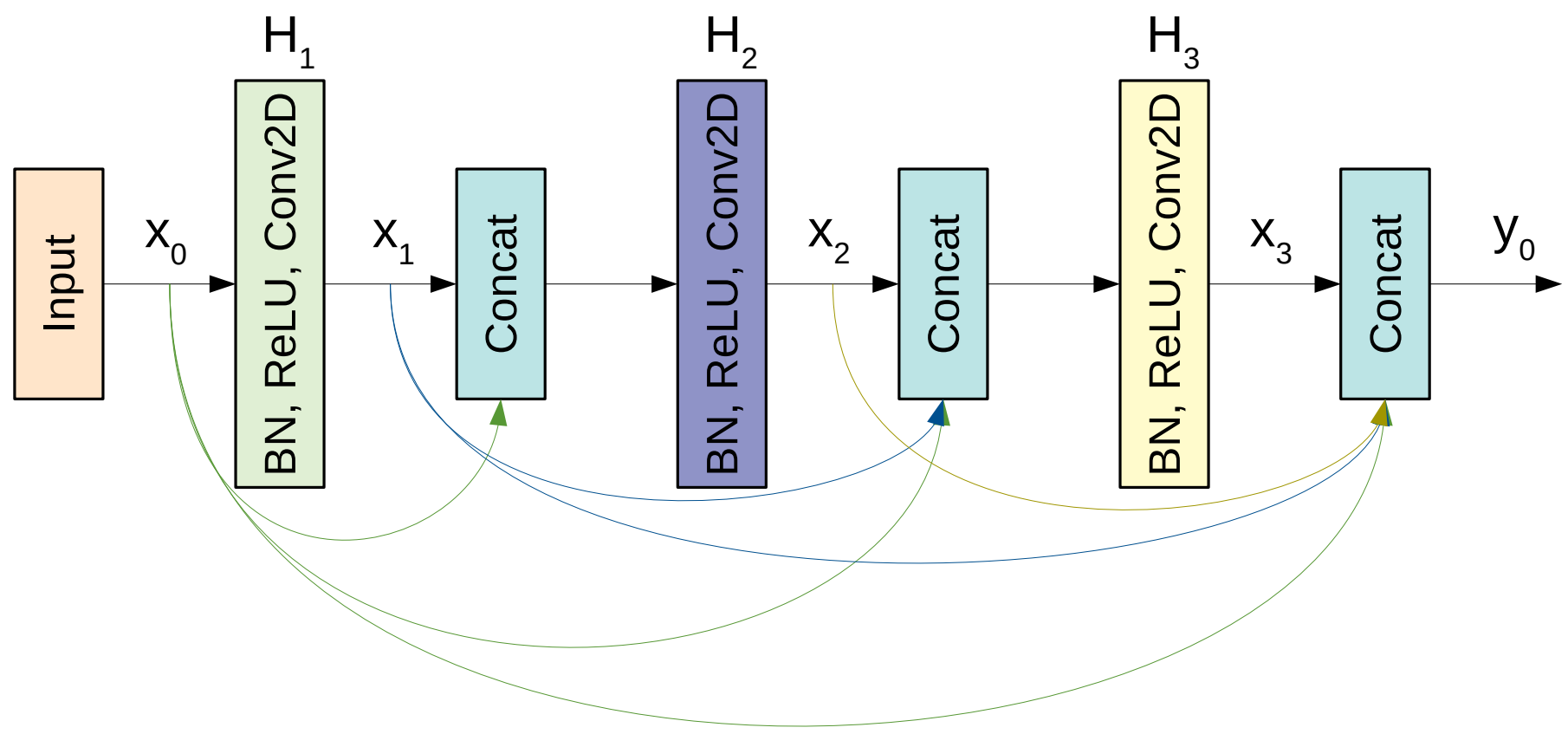

**Figure 3** **A schematic diagram of a 3-layer dense block used in the DenseNet architecture.**

of blocks as well as the input and output volumes are summarized in Table 3. In order to use CovidDenseNet with transfer learning, we implement the network in a three-step procedure. It involves downloading a pretrained DenseNet121, removing 2, 22 and 15 layers from the second, third and fourth dense block respectively, adding the adapter layers and then freezing the first convolutional layer, as well as the first dense block.

## METHODOLOGY

Figure 4 illustrates the entire process on how we conduct our experiments. We show that on the SARS-CoV-2 CT-scan dataset, which has three main classes. The dataset is split into training and test sets comprising 60% and 40% of the images from each class, respectively. Suitable preprocessing steps for normalization and data augmentation are carried out before training the models. The evaluation and testing are performed using various standard performance evaluation metrics. In order to reduce the variance of deep neural network models and improve the performance of the single models we consider the ensemble learning approach. An ensemble is built using several independently trained deep models and then combining the predictions from these models. At test time, each CT image is passed to the considered models where each model returns a vector of class scores known as the posterior probabilities. The resulting vectors of posterior probabilities from all models are then averaged and the actual prediction is given as the class with the highest probability. This is a commonly used ensembling approach for neural networks and is referred to as a committee of networks. More details about each step are described in the following subsections.

### Datasets

In order to evaluate our proposed models we used two benchmark CT image datasets, which are described below.

**Table 3 Our CovidDenseNet architecture for COVID-19 detection.** The network accepts an RGB-input of size 253 × 349 pixels.

| Layers | Output size | CovidDenseNet |
|---|---|---|
| Convolution | 129 × 177 | 7 × 7, 64, stride 2 |
| Pooling | 64 × 88 | 3 × 3 max pooling, stride 2 |
| Dense Block [1] | 64 × 88 | $\begin{bmatrix} 1 \times 1, conv \\ 3 \times 3, conv \end{bmatrix} \times 6$ |
| Transition Layer [1] | 64 × 88 | 1 × 1 conv |
| | 32 × 44 | 2 × 2 average pooling, stride 2 |
| Dense Block [2] | 32 × 44 | $\begin{bmatrix} 1 \times 1, conv \\ 3 \times 3, conv \end{bmatrix} \times 10$ |
| Adapter Layer [2] | 32 × 44 | 1 × 1 conv, 512 channels |
| Transition Layer [2] | 32 × 44 | 1 × 1 conv |
| | 16 × 22 | 2 × 2 average pooling, stride 2 |
| Dense Block [3] | 16 × 22 | $\begin{bmatrix} 1 \times 1, conv \\ 3 \times 3, conv \end{bmatrix} \times 2$ |
| Adapter Layer [3] | 16 × 22 | 1 × 1 conv, 1024 channels |
| Transition Layer [3] | 16 × 22 | 1 × 1 conv |
| | 8 × 11 | 2 × 2 average pooling, stride 2 |
| Dense Block [4] | 8 × 11 | $\begin{bmatrix} 1 \times 1, conv \\ 3 \times 3, conv \end{bmatrix} \times 1$ |
| Adapter Layer [4] (opt.) | 8 × 11 | 1 × 1 conv, 1024 channels |
| Classification layer | 1 × 1 | 8 × 11 adaptive average pool |
| | | fully connected, softmax |

**Table 4 Number of subjects and CT scans for each category in the SARS-CoV-2 CT-scan dataset.**

| Dataset | No. | COVID-19 | Healthy | Others | Total |
|---|---|---|---|---|---|
| SARS-CoV-2 CT-scan | subjects | 80 | 50 | 80 | 210 |
| | images | 2168 | 758 | 1247 | 4173 |
| COVID19-CT | subjects | 216 | – | 171 | 337 |
| | images | 349 | – | 397 | 746 |

### SARS-CoV-2 CT-scan dataset

The SARS-CoV-2 CT-scan dataset (*Soares et al., 2020*) is considered one of the largest CT scan datasets currently available for research that follows a patient-wise structure. The CT scans have been collected in public hospitals in Sao Paulo, Brazil, with a total of 4173 CT scans for 210 different subjects. The CT scans are distributed into three classes, namely COVID-19, Healthy, and Others. The exact number of patients and CT scans for each category is summarized in Table 4. As the dataset contains patients with other pulmonary diseases and the CT images have variable sizes, the dataset is challenging. Figure 5 shows 12 CT images from the SARS-CoV-2 CT-scan dataset, where the first row includes 4 COVID-19 images, the second row shows 4 images from the Healthy class, and the third row illustrates 4 images with other lung diseases from the Others class.

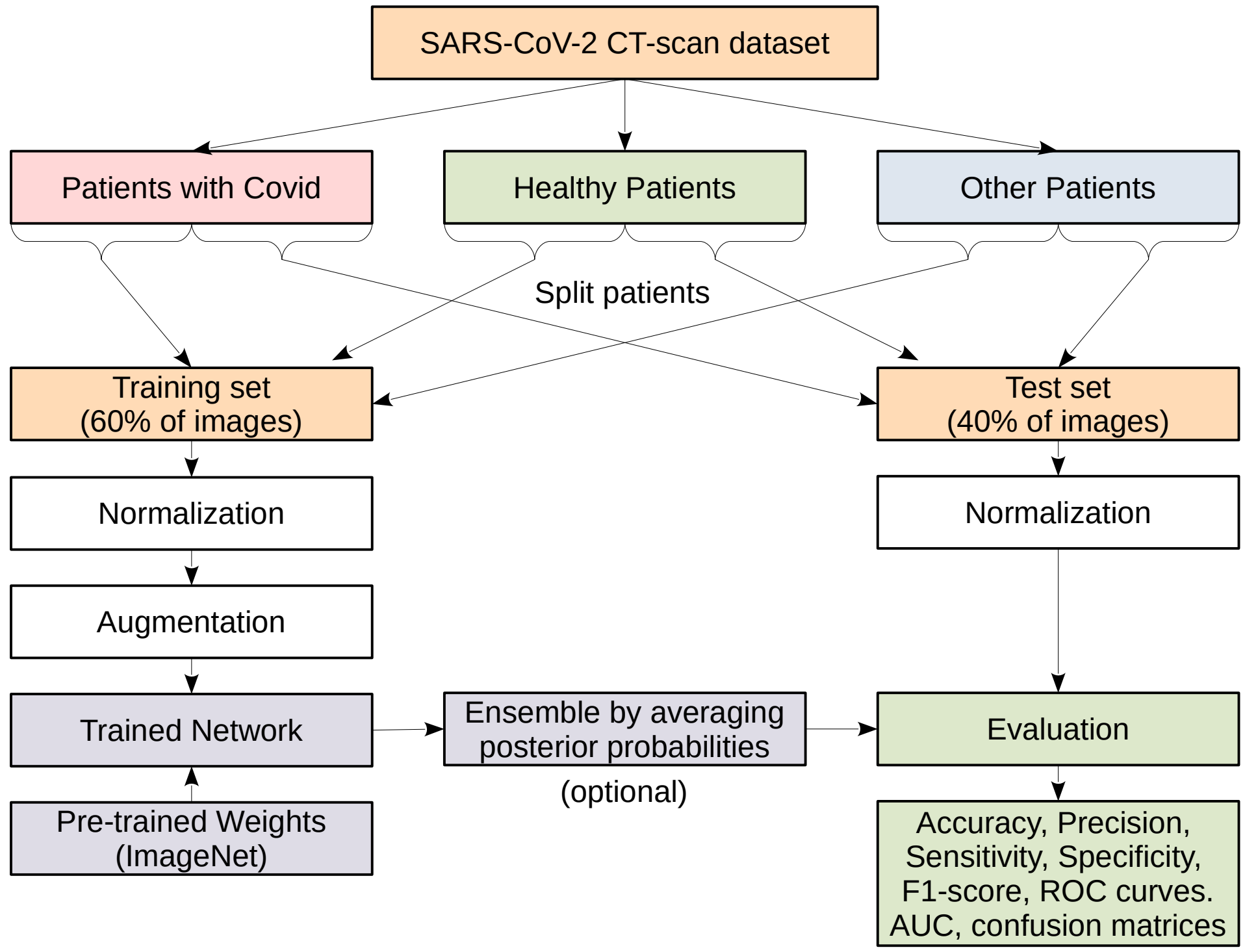

**Figure 4** **A flowchart representing the various steps in the process of models evaluation.**

### *COVID19-CT dataset*

The COVID19-CT dataset (*Zhao et al., 2020*) contains a total of 746 CT images. 349 CT images for patients with confirmed COVID-19 and 397 CTs for patients with other non-COVID-19 viral pneumonia. The CT images for COVID-19 were gathered from preprints on medRxiv and bioRxiv and they show various manifestations of COVID-19 pneumonia. The CTs for non-COVID19 cases were collected from different open sources. Due to the heterogeneity of sources, the images have different visual characteristics and varying sizes between $124 \times 153$ and $1485 \times 1853$, which make the dataset very challenging. Figure 6 shows sample CT images from the COVID19-CT dataset.

## Data preprocessing and splitting

Wide variations in the CT image sizes in the SARS-CoV-2 CT-scan and COVID19-CT datasets ask for a strategy to resize the images to a consistent input dimension for the network. The most frequently used approach to unify images with different aspect rations involves stretching, which can result in images that look unnatural or distorted. Therefore, we opt for a different procedure to preserve the aspect ratio by embedding the image into a fixed-sized canvas. We apply padding with the average color of the ImageNet dataset (*Deng et al., 2009*) when necessary to match the target shape. We empirically tried different input sizes and found that a canvas with a spatial dimension of $253 \times 349$ works best for CT images from the considered datasets and our architectures. Due to the limited amount of training data and the fact that deep neural networks require large amounts of data to

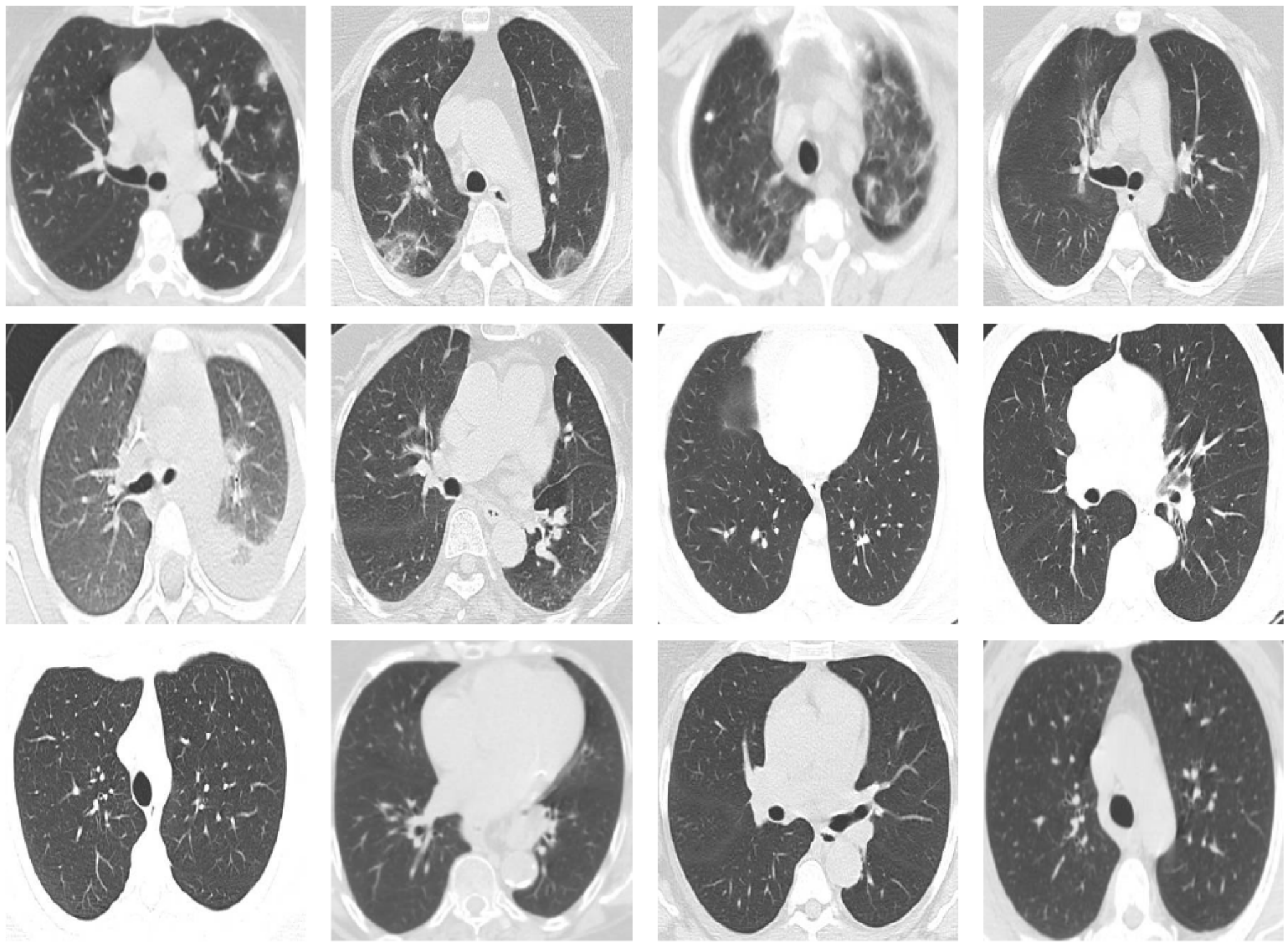

**Figure 5** **Sample CT images from the SARS-CoV-2 CT scan dataset.** The CTs represent four images of COVID-19 (first row), four images of Others class (second row), and four images from the Healthy class (third row).

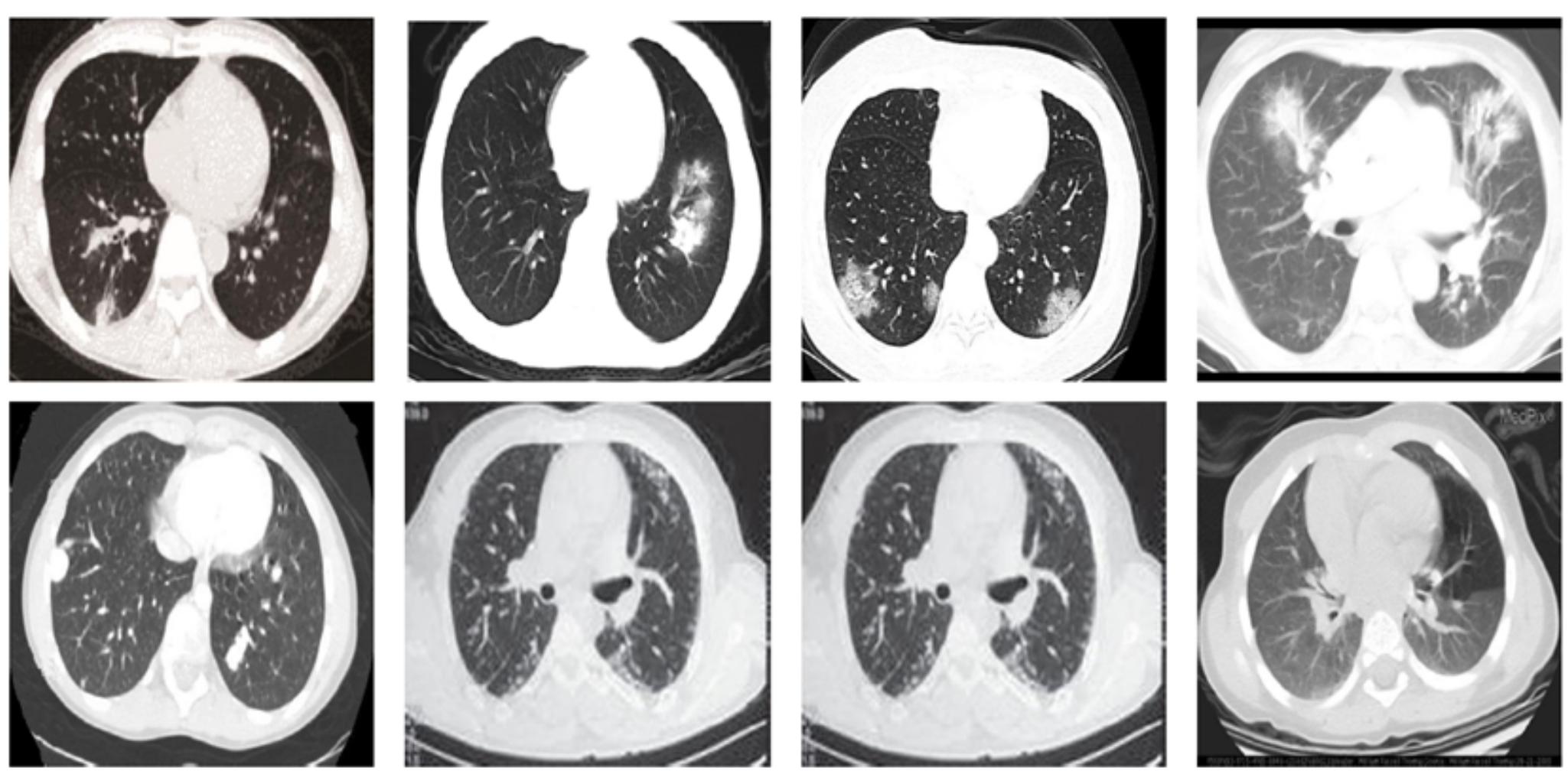

**Figure 6** **Sample CT images from the COVID19-CT dataset.** The first row shows four CT images of COVID-19 and the second row shows four CTs for non-COVID19 viral pneumonia.

optimize millions of parameters, we recompense the lack of data by implementing different augmentation steps to improve the network's ability to generalize. The augmentation steps include random rescaling, random cropping, Gaussian noise, brightness and contrast changes and random horizontal flipping. Finally, the images are normalized according to the mean and standard deviation of the ImageNet dataset.

To conduct our experiments we split the SARS-CoV-2 CT-scan dataset into training and test sets. We follow the subject-wise structure of the dataset, such that the two sets of persons in the training and test set are disjunct. Hence, it is assured, that we evaluate our models on unseen persons. However, the number of CT images per person vary. We divide the subjects into training and test sets such that the amounts of training and test images are 60% and 40%, respectively. The same ratio of persons is used for both scenarios of multi-class and binary classification tasks. Within one scenario we choose the same split for each architecture for the sake of consistency and comparability. However, for the COVID19-CT dataset we use the original split provided by *Zhao et al. (2020)*, where they split the dataset into 60% for training, 15% for validation, and 25% for testing and reporting results.

The class representations are imbalanced for the SARS-CoV-2-CT-scan and COVID19-CT datasets. We apply undersampling to make sure that our models are not biased towards class frequencies. In this procedure a new balanced subset is drawn from the training set every epoch. The images from this subset are augmented and used to update the model weights. Thus it is assured that the model was updated with the same amount of images of every class.

## Performance evaluation metrics

In order to evaluate the performance of our models we consider a set of standard quantitative evaluation metrics including:

$$Accuracy = (\mathrm{TP}+\mathrm{TN})/(\mathrm{TP}+\mathrm{TN}+\mathrm{FP}+\mathrm{FN}) \tag{1}$$

$$Precision = (\mathrm{TP})/(\mathrm{TP}+\mathrm{FP}) \tag{2}$$

$$Sensitivity = (\mathrm{TP})/(\mathrm{TP}+\mathrm{FN}) \tag{3}$$

$$Specificity = (\mathrm{TN})/(\mathrm{TN}+\mathrm{FP}) \tag{4}$$

$$F1-score = (2\times\mathrm{TP})/(2\times\mathrm{TP}+\mathrm{FP}+\mathrm{FN}) \tag{5}$$

where TP and TN refer to the total number of cases that are correctly classified as True Positives (TP) and True Negatives (TN), while FP and FN are the total number of cases that are incorrectly classified as False Positives (FP) and False Negatives (FN), respectively.

Table 5 **Confusion matrix for binary classification.**

| | Predicted values | |
|---|---|---|
| Actual values | True positive | False negative |
| | False positive | True negative |

We also report the macro average scores for the multi-class experiments to show the overall performance across the different classes of the dataset.

Models are difficult to compare when the performance assessment is based on a single evaluation metric only. Hence, we provide multiple evaluation metrics to enable a profound analysis. We plot the ROC curves to visualize the diagnostic ability of the models to differentiate between the different classes. We also compute the area under the ROC curve (AUC) for each model. The ROC curves show the trade-off between the true positive rate (sensitivity) and the false negative rate (1-specificity) at various threshold values. The AUC summarizes the ROC curve and measures the ability of a model to distinguish between the different classes. A high AUC value indicates better performance of the model at distinguishing between the classes.

We also provide the confusion matrix for detailed class-wise results. The confusion matrix clearly tells about the exact numbers of correctly detected positive and negative cases, as well as the type of error a model makes (see Table 5).

## Transfer learning

Transfer learning is a method in deep learning, which has become quite popular in the computer vision community because it might significantly boost recognition performance (*Alshazly et al., 2019b*). The idea bases on the transferability of network weights between related image recognition tasks and relies on the universal validity of the visual filters learned during training. Usually, transfer learning occurs in a two-step procedure. First, a model's weights are trained for one task on a dataset, which is typically large. Subsequently, a model is initialized with the weights to solve the actual task and often it is also fine-tuned.

As the size of the considered CT image datasets is still limited, we opt for transfer learning to benefit from the pretrained image filters. We initialize SqueezeNet (*Iandola et al., 2017*), VGG-16 (*Simonyan & Zisserman, 2015*), Inception-V3 (*Szegedy et al., 2016*), ResNet50 (*He et al., 2016*) and DenseNet121 (*Huang et al., 2017*) with weights, which have been optimized for the ImageNet dataset (*Russakovsky et al., 2015*). Parts of our proposed architectures exhibit compatible weight configurations such that we can initialize many weights with ResNet50 and DenseNet121 models that have been pretrained on ImageNet. In CovidResNet all weights are pretrained, but the last layer. In CovidDenseNet the adapter layers and the last layer are randomly initialized and all other weights are copied from the DenseNet121 model that was pretrained on ImageNet.

We empirically found that it is not necessary to adjust all weights to the COVID-19 detection problem. We assume that the filters from the first layers in a computer vision network provide somewhat generic filters that can be used for the SARS-CoV-2 CT-scan and COVID19-CT datasets. The idea is to reduce the risk of overfitting by lowering the amount of trained weights. Thus, we freeze the first convolutional layer and the first

**Table 6 Characteristics of CovidResNet and CovidDenseNet compared with the standard architectures.** The training time is given per epoch for the SARS-CoV-2 CT-scan dataset.

| Model | Network characteristics | | | | | |
|---|---|---|---|---|---|---|
| | Default input size | Custom input size | Layers | Total parameters (M) | Trainable parameters (M) | Training time (s) |
| SqueezeNet | 227 × 227 | 253 × 349 | 18 | 0.72 | 0.72 | 11.4 |
| VGG-16 | 227 × 227 | 253 × 349 | 16 | 134.28 | 134.28 | 27.4 |
| Inception-V3 | 299 × 299 | 253 × 349 | 48 | 21.79 | 21.79 | 16.9 |
| ResNet50 | 224 × 224 | 253 × 349 | 50 | 23.51 | 23.51 | 19 |
| DenseNet121 | 224 × 224 | 253 × 349 | 121 | 6.96 | 6.96 | 22 |
| CovidResNet | 253 × 349 | 253 × 349 | 29 | 9.84 | 9.61 | 11 |
| CovidDenseNet | 253 × 349 | 253 × 349 | 43 | 3.13 | 2.75 | 12 |

convolutional block of CovidResNet and only adapt the remaining weights. The first convolutional layer, the first dense block and the first transition layer of CovidDenseNet are also frozen. All weights in the models ResNet50 and DenseNet101 are fine-tuned to enable the comparison between standard models and our novel architectures together with our specifically designed fine tuning strategy. An overview of the CovidResNet and CovidDenseNet architectures can be seen in Fig. 1. The layers with frozen weights are highlighted in orange. The trainable layers are colored in blue. See Table 6 for important characteristics of the proposed CovidResNet and CovidDenseNet models compared with the standard architectures.

### Model training

All standard models are initialized using pretrained weights that have been optimized for the ImageNet dataset. The models are trained using the LAMB optimizer (*You et al., 2020*) with an initial learning rate of 0.0003 and cross-entropy loss. The standard models are trained for 100 epochs until convergence. Our proposed architectures and the SqueezeNet models need more epochs to converge and we stop training after 150 epochs. The reason for the higher number of epochs is the finding that it takes more iterations to train efficient networks that have fewer weights. The learning rate is step-wise reduced until a value of $10^{-6}$ is reached at the end of training. The batch size is 32. We also apply weight decay to regularize the training process. Optimization is performed within the PyTorch framework using an Nvidia GTX 1080 GPU.

## RESULTS AND DISCUSSION

This section presents and discusses the experimental results obtained by our proposed COVID-Nets architectures for COVID-19 detection. We begin our discussion by the obtained results for the SARS-CoV-2 CT-scan dataset, and then we discuss the obtained results for the COVID19-CT dataset.

## Results for SARS-CoV-2 CT-scan dataset

Here we report the obtained results on the SARS-CoV-2 CT-scan dataset for the three-class and the binary classification tasks. We also compare the performance of our models with state-of-the-art deep networks. First, we evaluate the ability of our models to differentiate patients with COVID-19, other non-COVID-19 viral lung infections, and non-infected healthy individuals. Second, we discuss the results obtained for all three possible binary classification scenarios from the same dataset.

### *Three-class Classification Results*

Table 7 provides the performance metrics, which are computed for each specific class, and the macro-average scores obtained by each model. Our proposed models achieve very promising results and outperform all standard fine-tuned models. Among the single network architectures, our CovidDenseNet model achieves the best overall performance with an average accuracy of 82.87%. Moreover, the model achieves the highest precision score of 95.76% for the COVID-19 class. Furthermore, the model achieves the best overall specificity score of 95.90% for COVID-19 class, which indicates its ability to designate most of the non-COVID-19 subjects as negative. However, the model obtains a sensitivity rate of 86.14% for the COVID-19 cases. The model also has a high sensitivity for the Others class. When considering the macro average scores for all evaluation metrics we observe that CovidDenseNet provides better performance compared to the other models. Similarly, our proposed CovidResNet model achieves better performance with respect to macro average precision, specificity and F1-score compared to all other models.

Based on our experimental results, which indicate superior performances for CovidResNet and CovidDenseNet, we considered these models for constructing ensembles for improving the overall diagnostic performance. The idea stems from the stochastic nature of deep networks where each network learns specific features and patterns. Building an ensemble of several independently trained networks and taking the unweighted average of their outputs can generate synergistic effects by exploiting the powerful feature extraction capability of each network (*Alshazly et al., 2019a*). Several ensemble combinations have been tested and we report the results of the best two ensembles in Table 7. We can see that in both cases, the ensemble models achieve better performance with respect to the macro average metrics compared to any individual network. Building an ensemble through a combination of our independently trained CovidResNet and CovidDenseNet models and their baselines increases the classification accuracy for all classes. When combining the prediction from CovidResNet and CovidDenseNet models (Ensemble 1) we notice a slight improvements over any of the single models in almost all evaluation metrics. Whereas combining CovidDenseNet and DenseNet121 models (Ensemble 2) improved the detection rate of CovidDenseNet for the COVID-19 class with 3%.

Figure 7 shows the confusion matrix for each of our proposed model and standard fine-tuned models, as well as the proposed ensembles that achieve the best performance.

**Table 7 Comparison of our models against state-of-the-art models for the multi-class problem.** The results are given in percentages and the best values are written in bold.

| Model | Class | Evaluation Metrics | | | | |
|---|---|---|---|---|---|---|
| | | Accuracy | Precision | Sensitivity | Specificity | F1-score |
| SqueezeNet | COVID-19 | | 86.99 | 87.30 | 85.95 | 87.15 |
| | Healthy | | 67.05 | 75.08 | 91.62 | 70.84 |
| | Others | | 68.35 | 62.83 | 87.74 | 65.47 |
| | Macro average | 77.78 | 74.13 | 75.06 | 88.44 | 74.49 |
| VGG-16 | COVID-19 | | 90.70 | 87.87 | 90.30 | 89.27 |
| | Healthy | | 64.66 | 79.94 | 90.08 | 71.49 |
| | Others | | 76.61 | 69.49 | 91.06 | 72.88 |
| | Macro average | 80.96 | 77.33 | 79.10 | 90.48 | 77.88 |
| Inception-V3 | COVID-19 | | 88.55 | 90.18 | 87.44 | 89.36 |
| | Healthy | | 66.95 | 76.05 | 91.48 | 71.21 |
| | Others | | 77.35 | 68.28 | 91.57 | 72.53 |
| | Macro average | 81.08 | 77.62 | 78.17 | 90.16 | 77.70 |
| ResNet50 | COVID-19 | | 87.49 | 91.22 | 85.95 | 89.32 |
| | Healthy | | 66.94 | 79.94 | 91.04 | 72.86 |
| | Others | | 82.16 | 66.06 | 93.96 | 73.24 |
| | Macro average | 81.68 | 78.86 | 79.07 | 90.31 | 78.47 |
| DenseNet121 | COVID-19 | | 89.83 | 89.72 | 89.05 | 89.77 |
| | Healthy | | 68.39 | 72.82 | 92.36 | 70.53 |
| | Others | | 75.21 | 72.32 | 89.96 | 73.74 |
| | Macro average | 81.44 | 77.81 | 78.29 | 90.46 | 78.02 |
| **Our CovidResNet** | COVID-19 | | 92.85 | 88.45 | 92.66 | 90.60 |
| | Healthy | | 66.67 | 77.67 | 91.18 | 71.75 |
| | Others | | 76.49 | 74.95 | 90.30 | 75.71 |
| | Macro average | 82.46 | 78.67 | 80.36 | 91.38 | 79.35 |
| **Our CovidDenseNet** | COVID-19 | | 95.76 | 86.14 | 95.90 | 90.70 |
| | Healthy | | 63.41 | 84.14 | 88.98 | 72.32 |
| | Others | | 78.59 | 76.36 | 91.23 | 77.46 |
| | Macro average | **82.87** | **79.25** | **82.22** | **92.04** | **80.16** |
| **Proposed Ensemble 1** | COVID-19 | | 94.03 | 87.30 | 94.03 | 90.54 |
| | Healthy | | 66.23 | 82.52 | 90.45 | 73.49 |
| | Others | | 76.36 | 76.36 | 91.23 | 77.46 |
| | Macro average | 83.17 | 79.62 | 82.06 | 91.90 | 80.49 |
| **Proposed Ensemble 2** | COVID-19 | | 93.24 | 89.15 | 93.03 | 91.15 |
| | Healthy | | 68.72 | 79.61 | 91.77 | 73.76 |
| | Others | | 79.13 | 77.37 | 91.40 | 78.24 |
| | Macro average | **83.89** | **80.36** | **82.04** | **92.07** | **81.05** |

By analyzing the confusion matrix we get insights on the class specific results achieved by each model with respect to the number of correctly classified and misclassified cases.

We also plot the ROC curves and compute the AUC to investigate the diagnostic accuracy of the proposed models for the multi-class problem in Fig. 8. Our CovidResNet

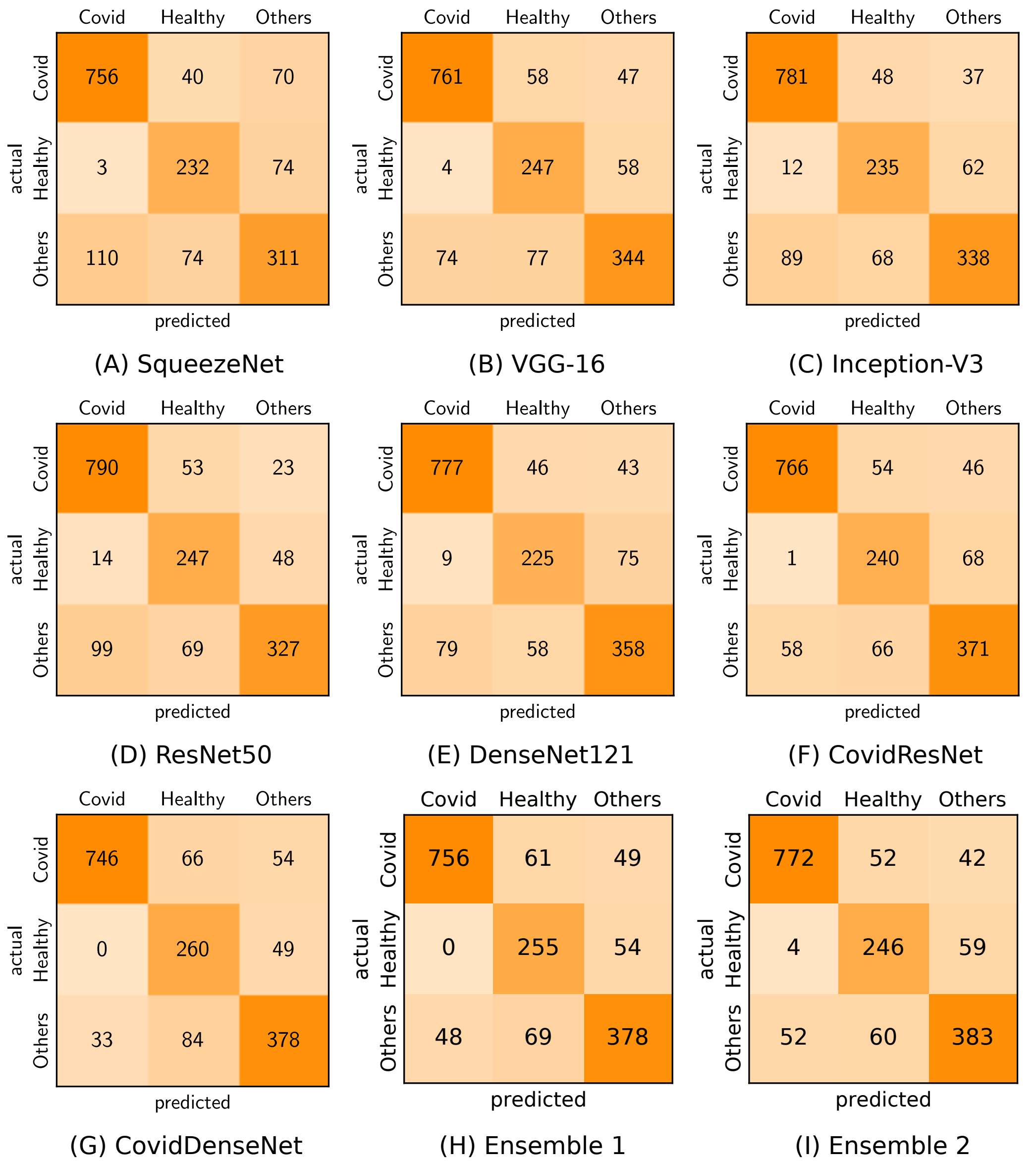

**Figure 7 Confusion matrices (A-I) generated by the different models for the three-class classification task.**

and CovidDenseNet models show superior performance and achieve higher AUC scores for the classes COVID-19 and Others, which indicates that our models detect COVID-19 and the other lung infections better than the deeper models of ResNet50 and DenseNet121. The AUC scores for the class Healthy is quite low as it has fewer number of subjects and CT images, which could be insufficient to learn discriminative features for separating this class from the other two classes. The superiority of the ensembles over single models is also reflected in the ROC curves and their corresponding AUC scores. When combining CovidDenseNet and CovidResNet, Ensemble 1, we notice an improvement in the AUC score for the all three classes within 2%. Similar results are achieved when we combine the

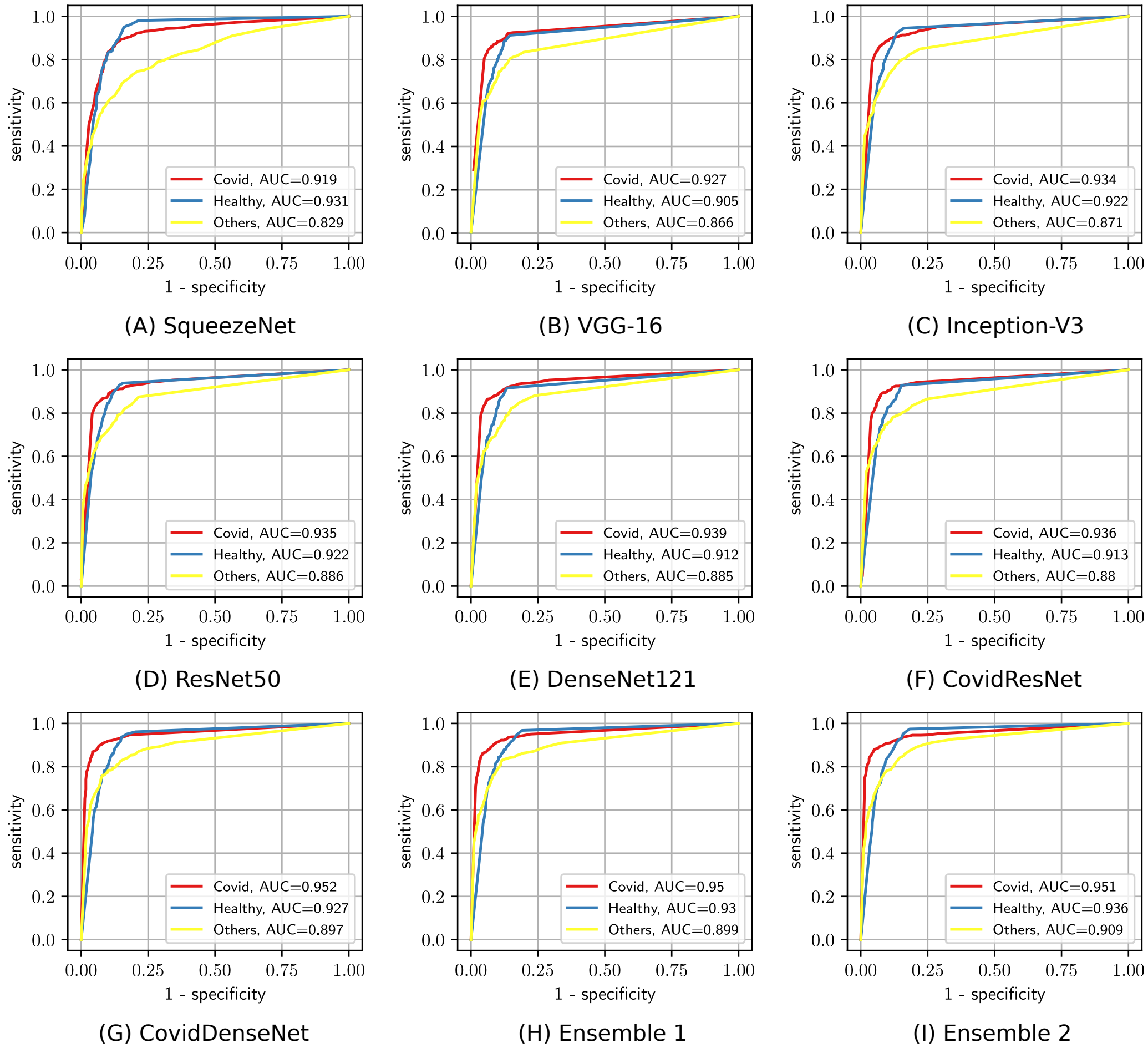

**Figure 8 The ROC curves and their AUC scores for the different models (A-I) showing their ability to differentiate between the three classes.**

CovidDenseNet and DenseNet121, Ensemble 2, even though the models were trained on the same training split of the used dataset.

### *Two-class classification results*

We tested our proposed architectures on binary classification tasks to validate their ability to distinguish between CT images of all possible classes, as well as to investigate the difficulty of these subtasks from the considered dataset. We investigate three experimental scenarios. First, we tested our models to differentiate patients with COVID-19 from healthy individuals (COVID-19 vs. Healthy). Then, we tested the models to distinguish COVID-19 cases from non-COVID-19 patients infected by other lung diseases (COVID-19 vs. Others). Finally, we tested our models to differentiate non-COVID-19 patients infected by other pulmonary diseases from healthy subjects (Others vs. Healthy). Table 8 presents the results obtained by each model under each of these scenarios.

In the first scenario (COVID-19 vs. Healthy) we used 866 CT images of COVID-19 and 309 CT images from the healthy class for testing. As we can see from Table 8 and

**Table 8** The obtained results under three binary classification scenarios.

| Task | Model | Evaluation Metrics | | | | |
|---|---|---|---|---|---|---|
| | | Accuracy | Precision | Sensitivity | Specificity | F1-score |
| COVID-19 vs. Healthy | SqueezeNet | 92.85 | 95.78 | **94.46** | 88.35 | 95.12 |
| | VGG-16 | 93.96 | 97.95 | 93.76 | 94.50 | 95.81 |
| | Inception-V3 | 93.19 | 96.90 | 93.76 | 91.59 | 95.30 |
| | ResNet50 | **93.96** | 98.18 | 93.53 | 95.15 | **95.80** |
| | DenseNet121 | 93.53 | 97.14 | 94.00 | 92.23 | 95.54 |
| | CovidResNet | 93.87 | **99.13** | 92.49 | **97.73** | 95.70 |
| | CovidDenseNet | 93.11 | 97.01 | 93.53 | 91.91 | 95.24 |
| COVID-19 vs. Others | SqueezeNet | 81.25 | 86.88 | 83.372 | 77.43 | 85.09 |
| | VGG-16 | 85.24 | 88.38 | 88.68 | 79.09 | 88.53 |
| | InceptionV3 | 81.76 | 84.68 | 87.41 | 71.63 | 86.02 |
| | ResNet50 | 83.77 | 85.59 | **89.84** | 72.88 | 87.66 |
| | DenseNet121 | 83.10 | 85.21 | 89.15 | 72.26 | 87.13 |
| | CovidResNet | 85.10 | 90.80 | 85.45 | 84.47 | 88.04 |
| | CovidDenseNet | **86.88** | **91.76** | 87.41 | **85.92** | **89.53** |
| Others vs. Healthy | SqueezeNet | 86.15 | **91.04** | 86.44 | **85.67** | 88.68 |
| | VGG-16 | 85.13 | 87.78 | 88.66 | 79.18 | 88.22 |
| | Inception-V3 | 85.13 | 86.32 | **90.69** | 75.77 | 88.45 |
| | ResNet50 | 85.64 | 90.97 | 85.63 | **85.67** | 88.22 |
| | DenseNet121 | 83.35 | 85.52 | 88.46 | 74.74 | 86.97 |
| | CovidResNet | **86.40** | 88.32 | 90.28 | 79.86 | **89.29** |
| | CovidDenseNet | 83.61 | 84.89 | 89.88 | 73.04 | 87.32 |

**Note.**
The results are given in percentages and the best values are written in bold.

under this scenario, the seven models achieve very competitive performance with accuracy above 93% and F1-score above 95%. The models also achieve high precision values above 96%, where our proposed CovidResNet model achieves the highest precision score of 99.13%, indicating that almost all the predicted subjects as COVID-19 are correct and only 7 out of 309 healthy CT images were incorrectly classified as COVID-19 positive. CovidResNet also attains the highest specificity score of 97.73%, which indicates its ability to correctly identify 302 out of 309 normal CT images as COVID-19 negative. However, CovidResNet has a lower sensitivity rate compared to other models. The model is able to correctly detect 92.49% of COVID-19 cases and 65 COVID-19 CTs were incorrectly detected as non-COVID-19 (false negatives). Nevertheless, this high false negative rate is a common problem among all the tested models and can be attributed to two main reasons. First, in some cases, patients with COVID-19 may show normal chest CT findings at the early days of infection, and therefore it is hard to exclude all COVID-19 cases based only on the chest CT predictive results. Second, the findings on CTs can be very tiny and can barely be detected by the models, as the CT images of COVID-19 patients may manifest different imaging characteristics such as specific patterns progressively with time based on

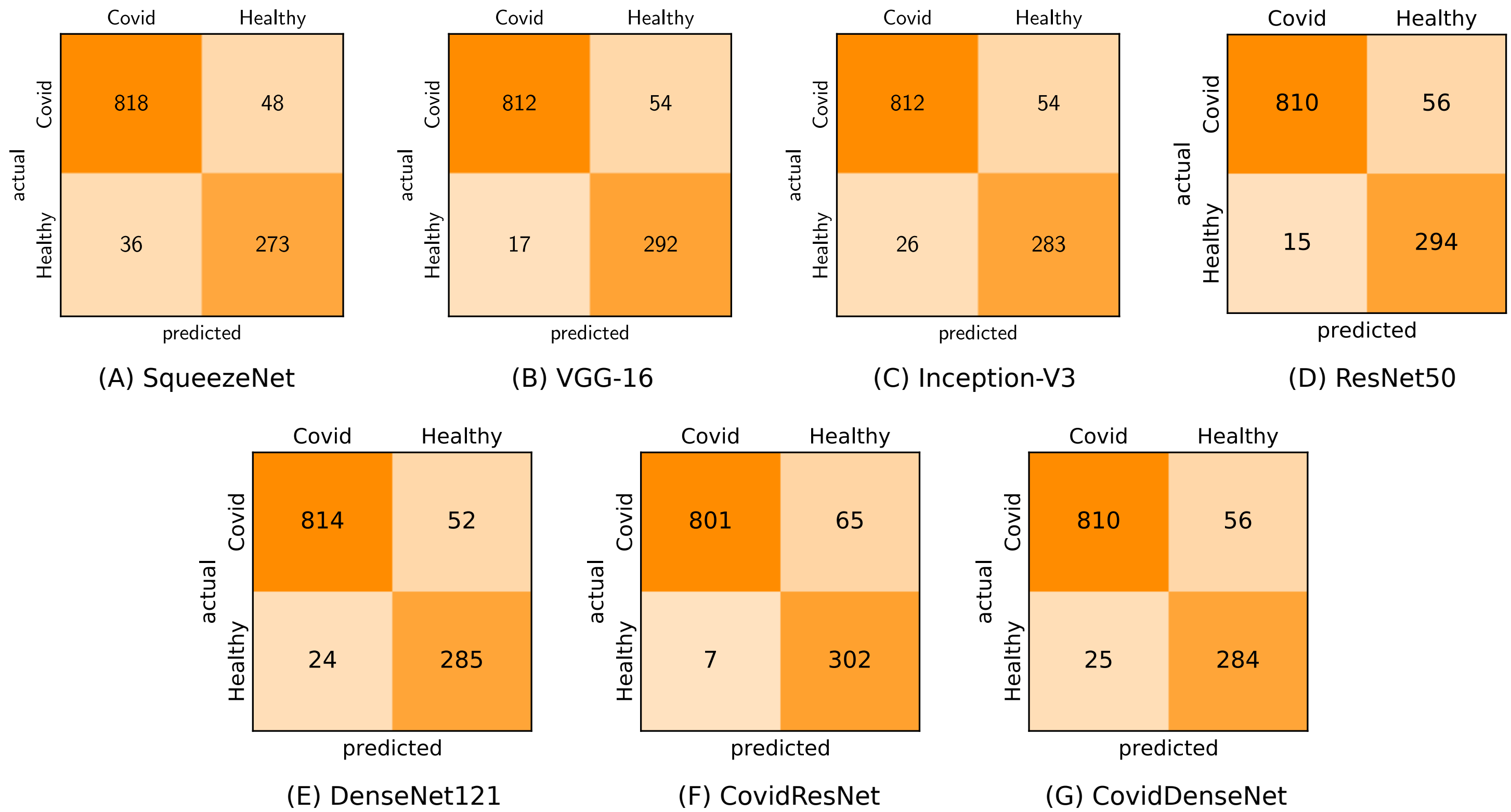

**Figure 9** **Confusion matrices (A-G) generated by all models for COVID-19 vs. healthy classification.**

the severity of the infection. For a detailed class-wise results, the confusion matrix for each specific model under the considered scenario is presented in Fig. 9.

Figure 10 shows the ROC curves for all evaluated models. Looking at the ROC curves and the AUC scores we can see that all tested models perform on a similar level. The ROC curves look identical and the AUC scores vary within a range less than 1%, with ResNet50 achieving a slightly higher AUC sore of 97.2%.

In the second scenario (COVID-19 vs. Others) we investigated the effectiveness of our models in differentiating the CTs of COVID-19 from others with viral lung infections. It is worth mentioning that this is a challenging task due to the potential overlap of findings on CT images between COVID-19 and the other non-COVID-19 viral infections. The obtained results in Table 8 clearly show lower performance with respect to all evaluation metrics compared to the obtained results in the first scenario. Nevertheless, our proposed CovidResNet and CovidDenseNet models achieve higher accuracy values compared with the standard models, where our CovidDenseNet model attains an accuracy of 86.88%. Our proposed models also achieve much better results with respect to precision, specificity, and F1-score values. Our CovidDenseNet model achieves the highest precision score of 91.76% indicating its ability to correctly identify CTs with COVID-19. Only 68 out of 483 CT images from the Others class are incorrectly classified as COVID-19 (false positives). It is also worth noting that our CovidResNet and CovidDenseNet models achieve much

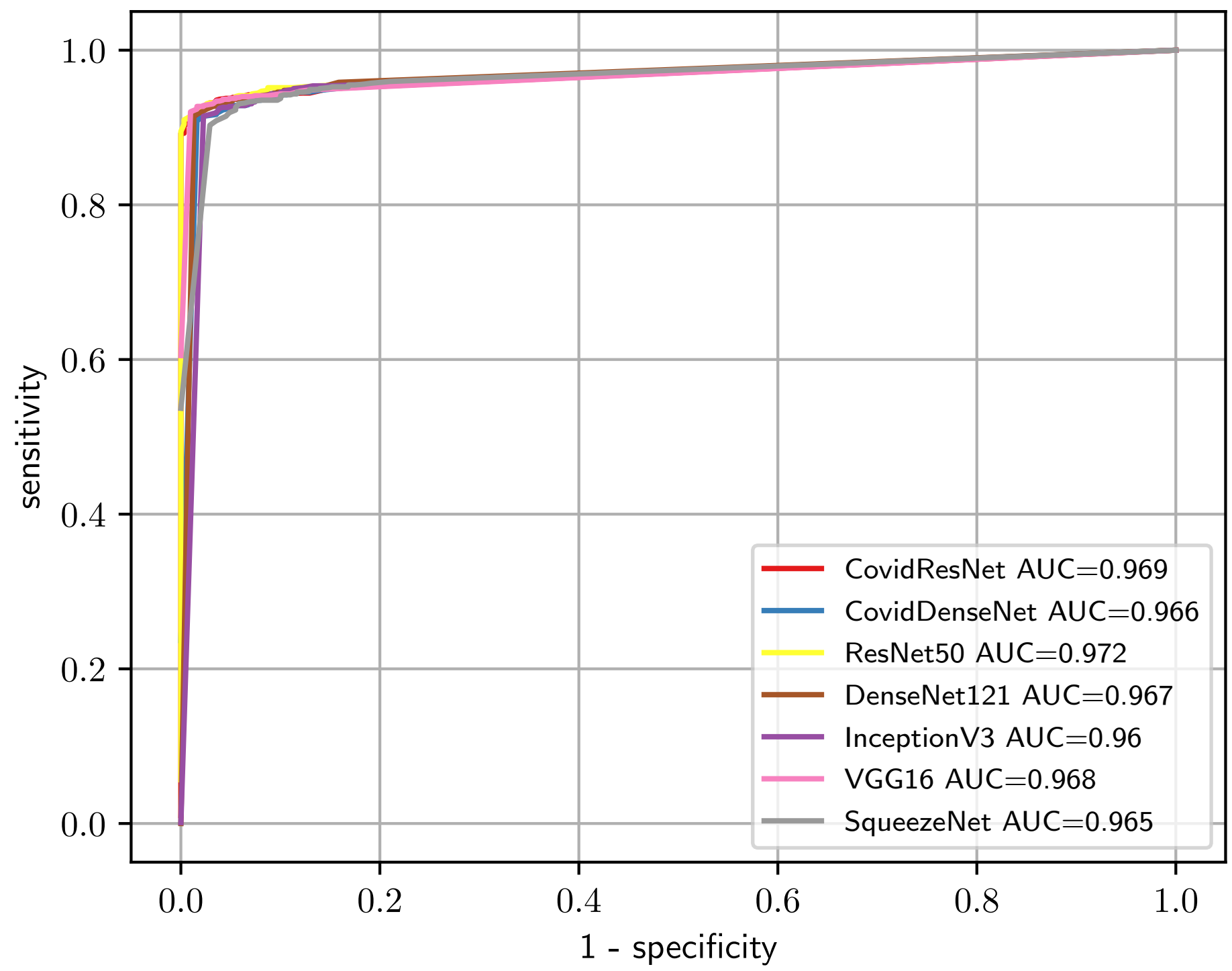

**Figure 10 Comparison of the predictive performance for CovidResNet and CovidDenseNet and the standard models for COVID-19 vs. healthy classification. The ROC curves and AUC scores show the competitive performance for all models.**

higher specificity rates above 85% outperforming the standard models with 12%. The lower specificity of the standard models may stem from the difficulty to distinguish the CT findings of COVID-19 from findings of other non-COVID-19 viral diseases. On the contrary, our CovidDenseNet model correctly detected 415 out of 483 CT images as other lung diseases. However, our models show slightly lower sensitivity rates compared to the other models due to more false negatives. Nevertheless, a high false negative rate is a common issue for all the tested models due to the potential overlap of the imaging findings. By investigating the confusion matrix we get a detailed class-wise analysis. Figure 11 shows the confusion matrix for each model and what type of error each specific model makes.

We also compare the performance of the different models under this scenario by plotting the ROC curve and computing the AUC score for each model. Figure 12 shows the ROC curves, where we clearly see that our CovidResNet and CovidDenseNet model are superior to the other models as their ROC curves are closer to the top-left corner and they achieve higher AUC values. The highest AUC score of 92.4% is achieved by our CovidDenseNet model exceeding its deeper counterpart DenseNet121 model with more than 5%.

In our third scenario (Others vs. Healthy) we tested the ability of our architectures to differentiate patients infected with other pulmonary diseases and non-infected healthy individuals. While our main objective in this work is to develop architectures to differentiate patients with COVID-19 from other non-COVID-19 viral infections as well as healthy

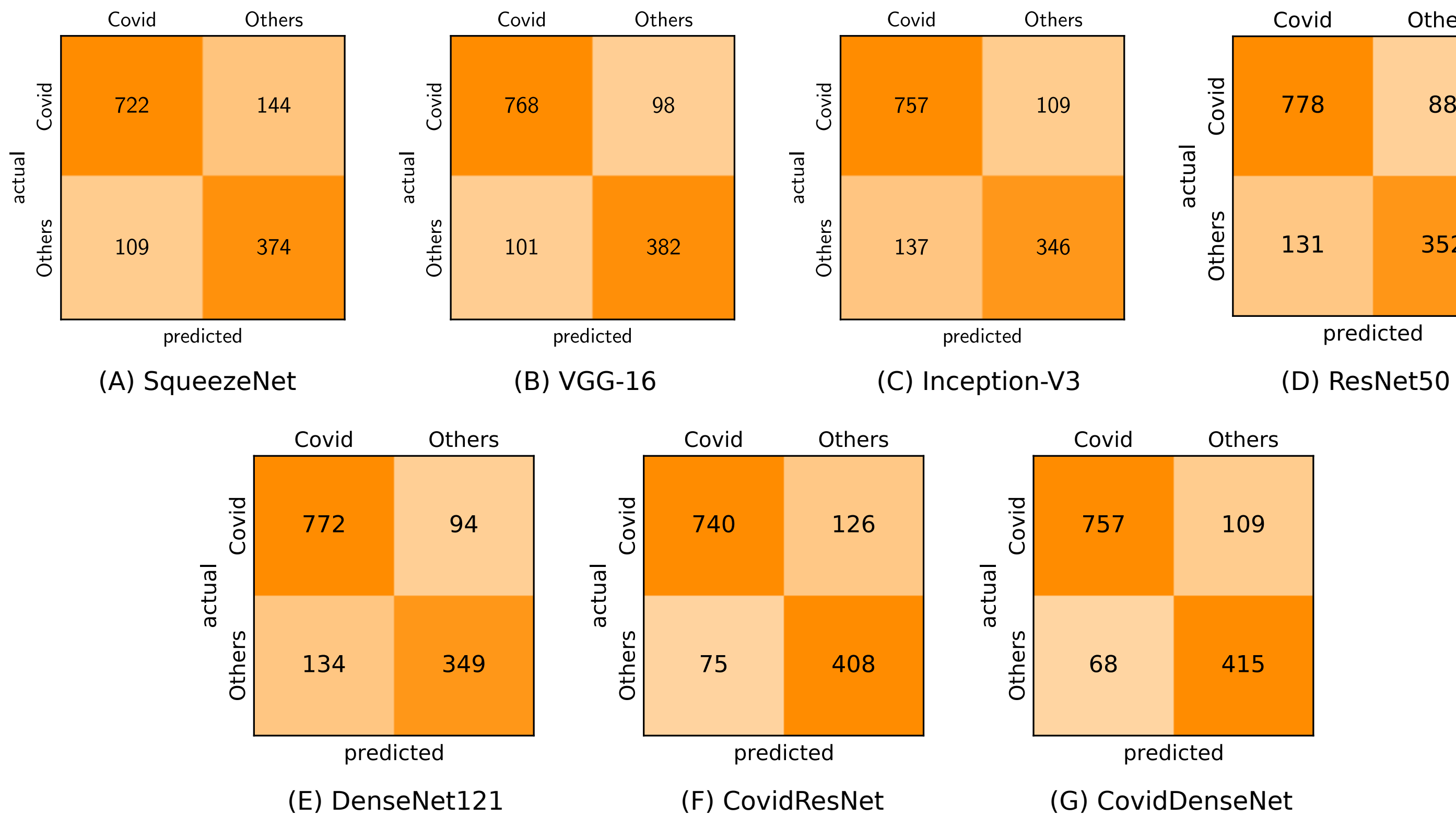

**Figure 11** **Confusion matrices (A-G) for COVID-19 vs. others classification for all tested models.**

subjects, we report our results under this scenario for the sake of completeness. In our experiments we treat people infected by other viral infections as the positive class and the healthy individuals as the negative class. Under this scenario, our CovidResNet model achieves the best overall performance with 86.40% accuracy. The model also achieves the highest F1-score of 89.29%. With a sensitivity score of 90.28% CovidResNet indicates its ability to detect above 90% of the infected cases. Figure 13 shows the confusion matrix for each of the tested models. We can observe that all the models have high false positive rates under this scenario compared with the first scenario (COVID-19 vs. Healthy). A possible reason is that we have more CT images in the COVID-19 class to learn fairly discriminative features, whereas the limited amount of CT scans for the Others class makes it difficult to distinguish them from non-infected or normal CT images. Therefore, we need to collect more CT images for both classes to reduce the false positive as well as the false negative rates.

Figure 14 presents the ROC curves and their corresponding AUC scores for all tested models. Again, our proposed models show superior performance compared with other deeper models. Our CovidResNet model achieves a high AUC score of 92.8% comparable to the best model (SqueezeNet) and its ROC curve appears closer to the top-left corner. Our CovidDenseNet model also has a comparable performance to the standard models

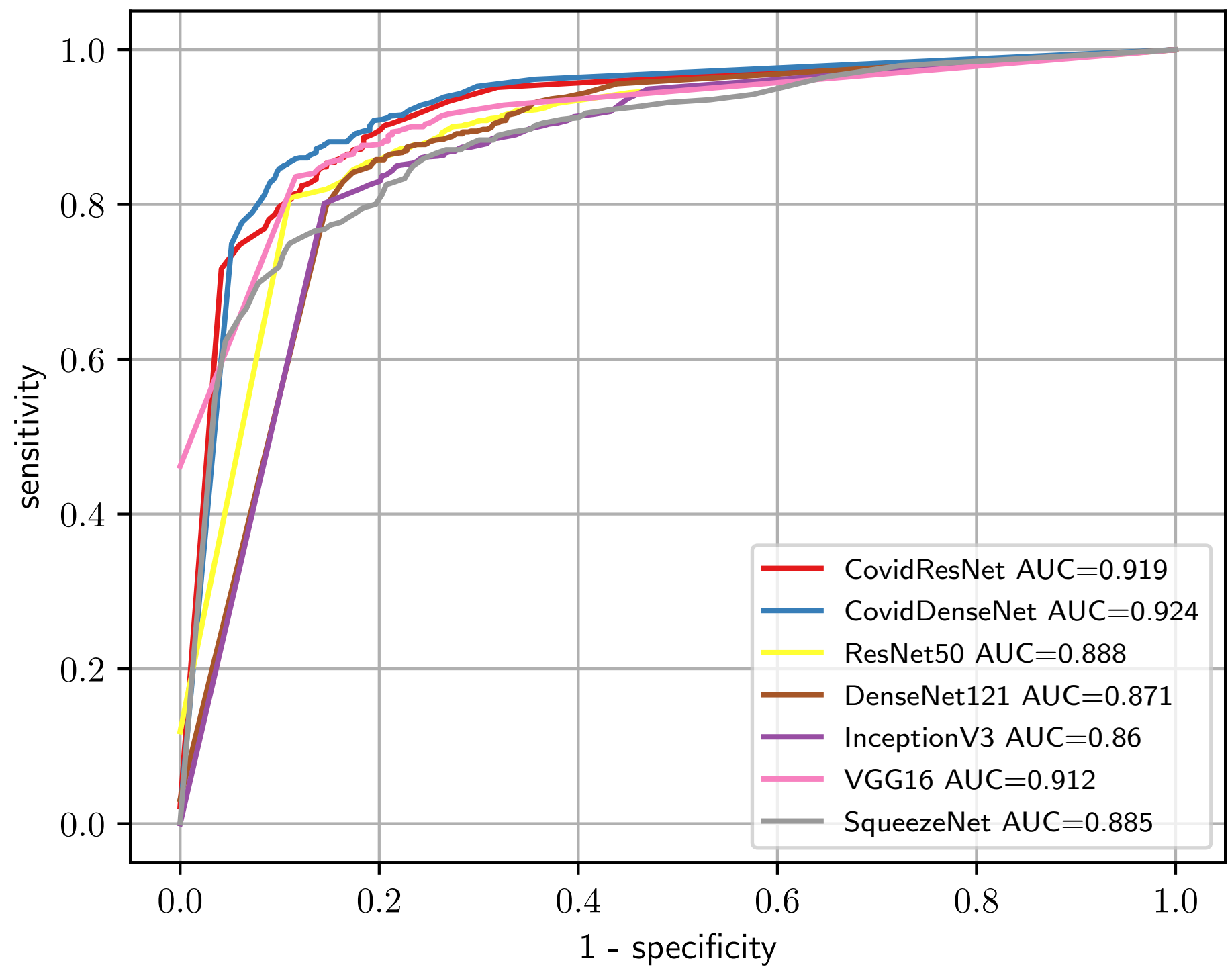

**Figure 12** **Predictive performance of our proposed CovidResNet and CovidDenseNet models against the standard models for COVID-19 vs. others classification. The ROC curves show powerful a predictive power for the CovidDenseNet model.**

with AUC score of 89.8%, which indicates approximately 2% improvement compared to its deeper DenseNet121 model.

## Results for the COVID19-CT dataset

To validate the effectiveness of our proposed architectures we use the COVID19-CT dataset, which shares similar visual characteristics with the SARS-CoV-2 CT-scan dataset and is also available in a subject-wise structure. We consider the binary classification scenario of differentiating COVID-19 cases from other non-COVID viral pneumonia. We report our results based on the test split provided in *Zhao et al. (2020)* where 98 COVID-19 and 105 non-COVID CT images are used. Table 9 shows the performance of our CovidResNet and CovidDenseNet models compared with the standard models.

From the table, we observe the competitive performance of our proposed models in terms of accuracy, sensitivity, and F1-score. Moreover, our models achieved much better results compared with the reported results from the literature. Looking closely to the results in Table 9 we can see a similarity with the second scenario in Table 8 when differentiating COVID-19 and other non-COVID CT images. Under both scenarios our CovidDenseNet model achieved the best performance, which indicates the effectiveness of our models to work on different datasets and to differentiate COVID-19 from other non-COVID cases.

In order to give a holistic view on how well our proposed models perform on the COVID19-CT dataset and to show what types of errors they are making, we present the

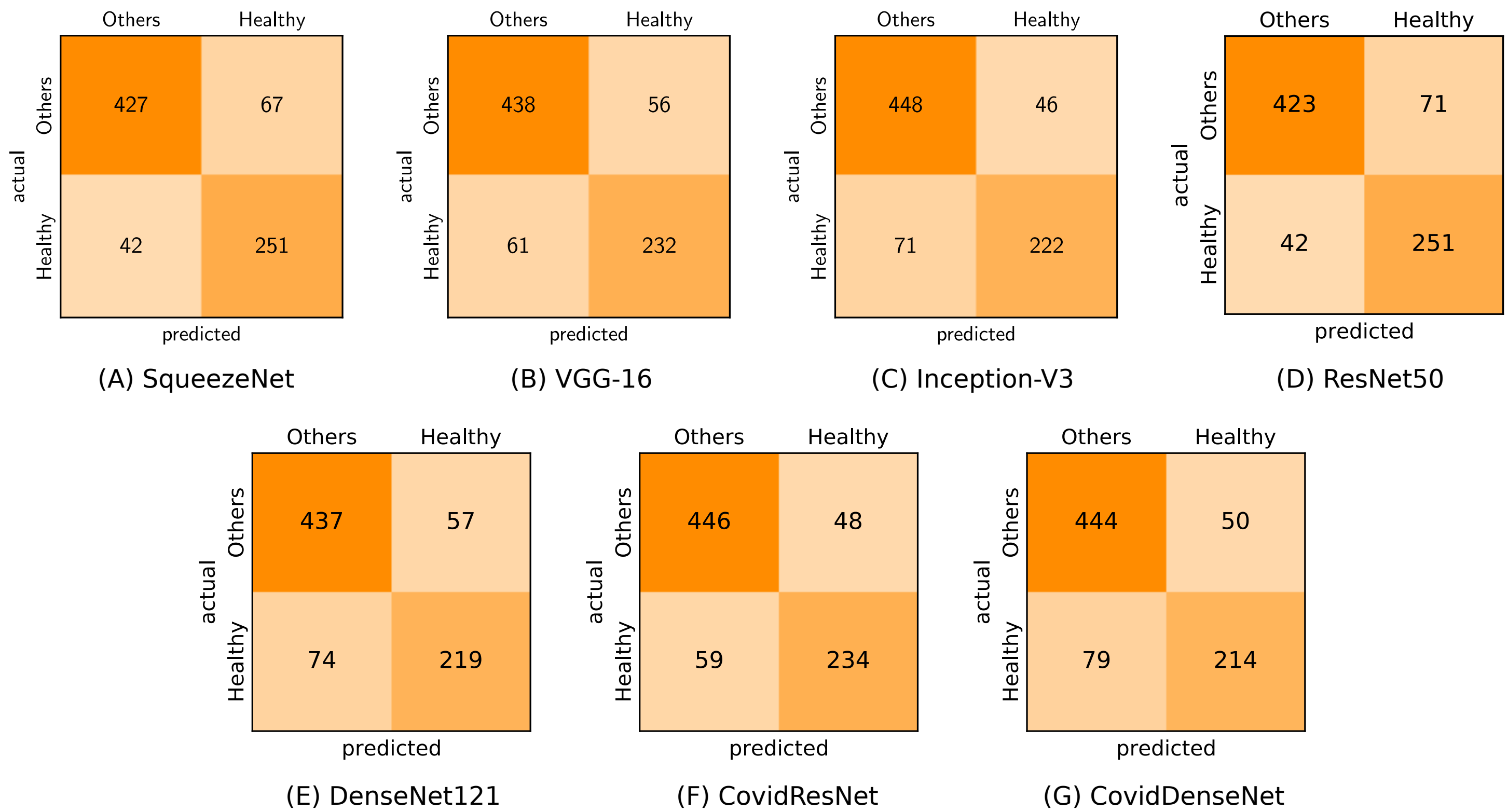

**Figure 13** **Confusion matrices (A-G) generated by all tested models for Others vs. healthy classification.**

confusion matrices for each model in Fig. 15. We also validate the ability of our models to distinguish between classes by plotting the ROC curves and computing the corresponding AUC for each model. Figure 16 depicts the ROC curves for the different models. We can see that our proposed models have the superior performance over the standard models. Our CovidDenseNet achieved the highest AUC score of 87.5% and its ROC curve is higher than the ROC curves for other models. Therefore, we can say that our CovidDenseNet model performed better than all models in detecting the positive class (i.e., COVID-19) in the COVID19-CT dataset.

## CONCLUSION

We proposed two deep CNN architectures (CovidResNet and CovidDenseNet) for the automated detection of COVID-19 using chest CT scans. The models were developed and validated on two benchmark CT image datasets. We also presented the first experimental study on the large-scale multi-class SARS-CoV-2 CT-scan dataset, which has more than 4000 CT scans. Extensive experiments have been conducted to evaluate our models in the multi-class and binary classification tasks from the SARS-CoV-2 CT-scan dataset. First, we trained our models to differentiate COVID-19 cases from other non-COVID-19 infections as well as from healthy subjects. Experimental results show the effectiveness of the proposed architectures to achieve better results compared with the standard architectures, while being more computationally efficient. Second, we conducted three binary classification

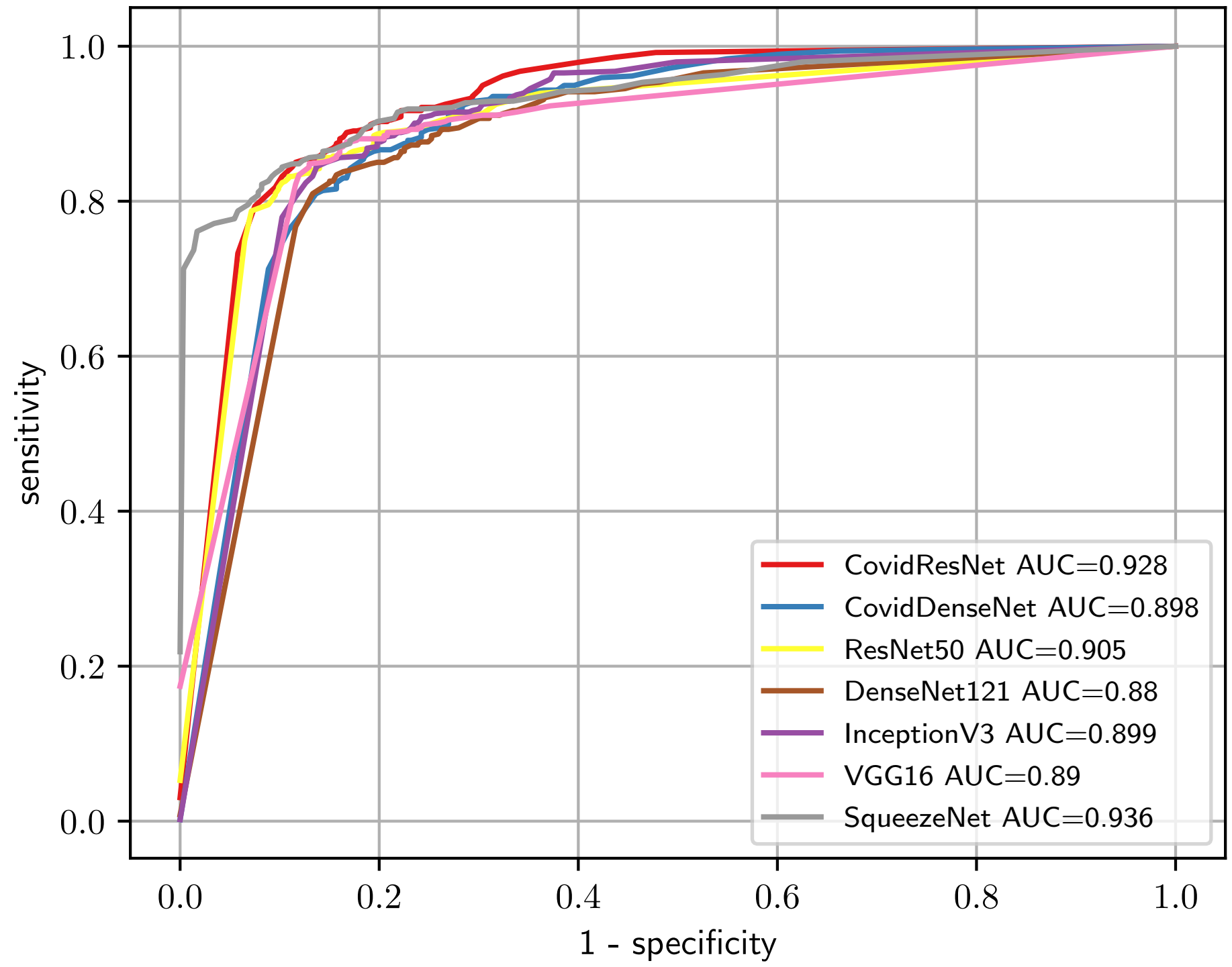

**Figure 14** **Comparison of CovidResNet and CovidDenseNet against the standard models using the ROC curves and AUC scores for the Others vs. healthy classification task.**

**Table 9** **Performance comparison of our proposed networks against all standard models and previous work on the COVID19-CT dataset.**

| Task | Model | Evaluation metrics | | | | |
|---|---|---|---|---|---|---|
| | | Accuracy | Precision | Sensitivity | Specificity | F1-score |
| COVID vs. Non-COVID | SqueezeNet | 73.89 | 79.22 | 62.24 | 84.76 | 69.70 |
| | VGG-16 | 80.79 | 81.05 | 78.57 | 82.85 | 79.79 |
| | Inception-V3 | 79.80 | 78.21 | 80.61 | 79.05 | 79.40 |
| | ResNet50 | **81.77** | 83.52 | 77.55 | 85.71 | 80.42 |
| | DenseNet121 | 81.28 | **84.09** | 75.51 | **86.67** | 79.57 |
| | CovidResNet | 81.28 | 79.41 | 82.65 | 80.00 | 81.00 |
| | CovidDenseNet | **81.77** | 79.05 | **84.69** | 79.05 | **81.77** |
| Previous Work | ResNet50 (*He et al., 2020*) | 69.0 | – | – | – | 72.0 |
| | DenseNet121 (*He et al., 2020*) | 76.0 | – | – | – | 77.0 |
| | CRNet (*He et al., 2020*) | 72.0 | – | – | – | 76.0 |
| | Redesigned COVID-Net (*Wang, Liu & Dou, 2020*) | 79.0 | – | – | – | 79.0 |
| | M-Inception (*Wang et al., 2021b*) | 81.0 | – | – | – | 82.0 |
| | Evidential Covid-Net (*Huang, Ruan & Denoeux, 2021*) | 81.0 | – | – | – | 81.0 |

**Note.**
The results are given in percentages and the best values are written in bold.

scenarios to differentiate COVID-19 from healthy individuals, COVID-19 from other non-COVID-19 patients, and non-COVID-19 viral infections from non-infected healthy

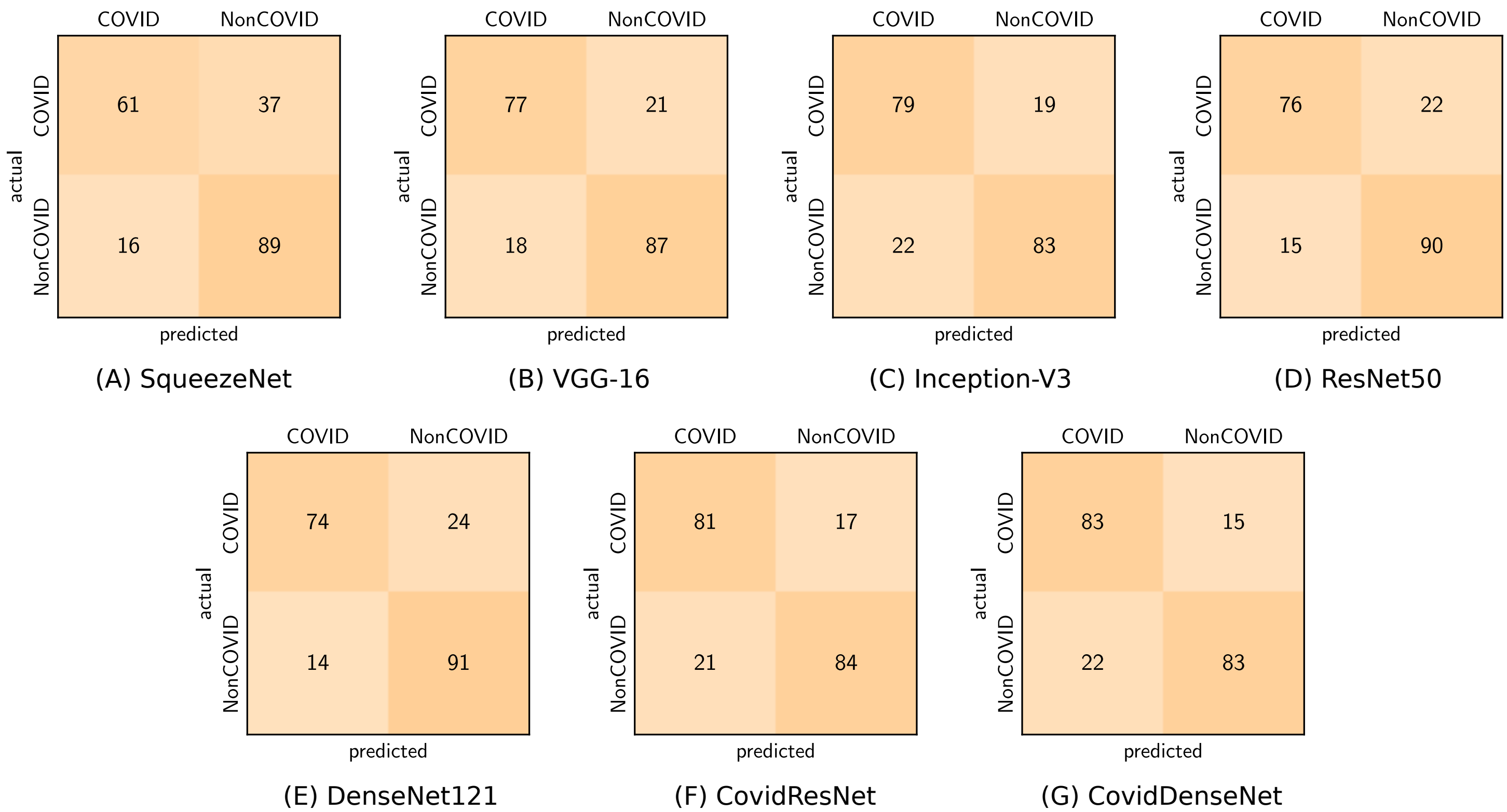

**Figure 15** **Confusion matrices (A-G) obtained by the different models for COVID vs. non-COVID classification from the COVID19-CT dataset.**

subjects. The obtained results revealed the superior performance of our proposed models over the baseline models. Finally, we tested our models on the COVID19-CT dataset to differentiate COVID-19 patients from others with non-COVID-19 viral infection. Our CovidDenseNet model achieved the best overall detection performance exceeding the state-of-the-art models.

To the best of our knowledge, this is the first experimental study on the SARS-CoV-2 CT-scan dataset that considers subject-wise splits for training and testing. Therefore, our models and results can be used as a baseline benchmark for any future experiments conducted on this dataset. Although our experimental results are promising, there is still room for improvement. We assume that experiments conducted on even larger datasets of CT scans will improve the diagnostic accuracy and provide a more reliable estimation of the models' performance. Collecting more CT scans and subjects for all classes and particularly the Healthy and Others categories can further improve the diagnostic performance of the proposed models.

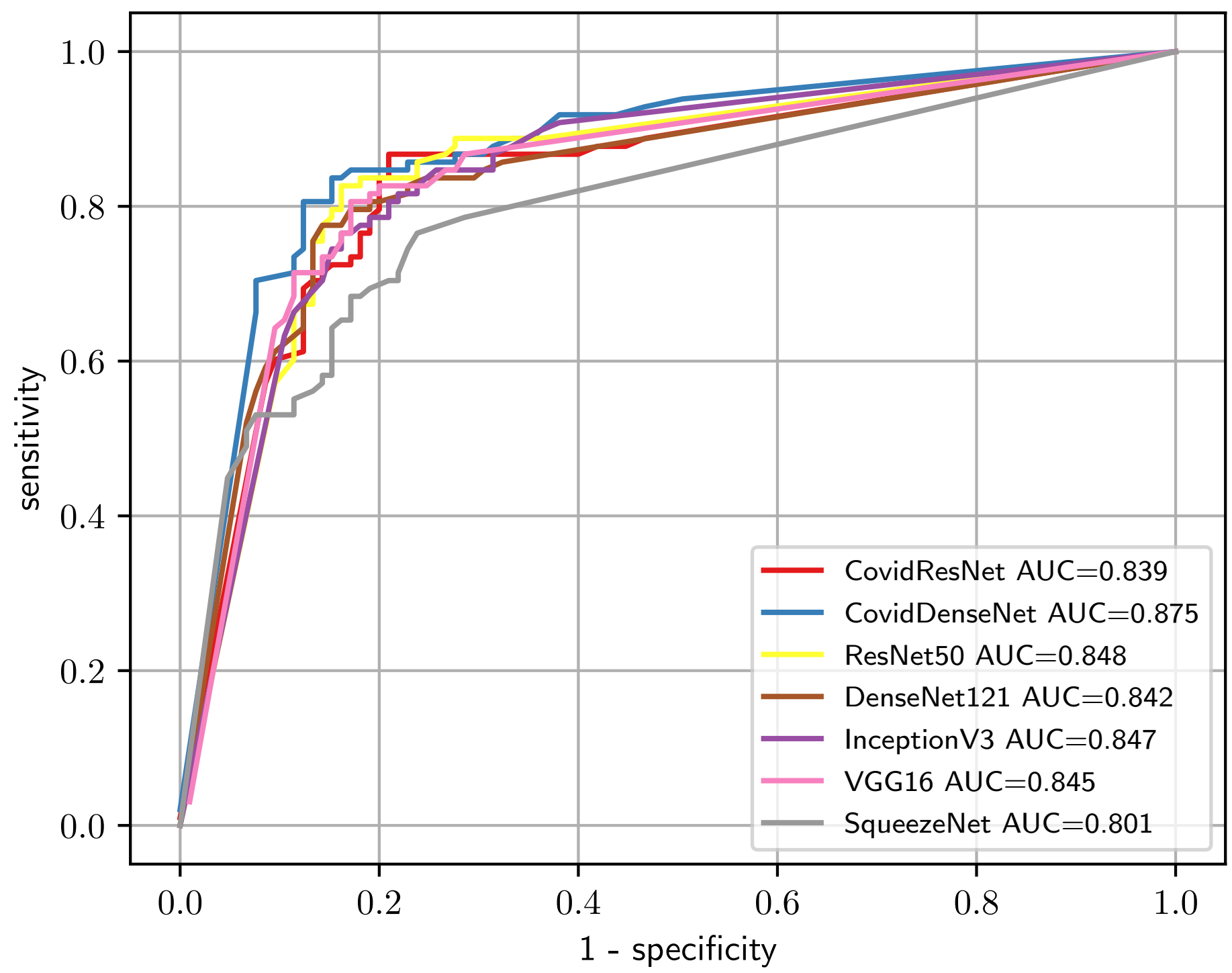

**Figure 16** **Comparison of the diagnostic ability of our CovidResNet and CovidDenseNet with the standard models to distinguish COVID-19 and non-COVID class in the COVID19-CT dataset.**

## Funding

Hammam Alshazly and Mohamed Abdalla received funding through the Research Group Program under grant number RGP.1/3/42 from the Deanship of Scientific Research at King Khalid University. The work of Christoph Linse was supported by the Bundesministeriums für Wirtschaft und Energie (BMWi) through the Mittelstand 4.0-Kompetenzzentrum Kiel Project. The funders had no role in study design, data collection and analysis, decision to publish, or preparation of the manuscript.

## Grant Disclosures

The following grant information was disclosed by the authors:
Deanship of Scientific Research at King Khalid University: RGP.1/3/42.
Bundesministeriums für Wirtschaft und Energie (BMWi) through the Mittelstand 4.0-Kompetenzzentrum Kiel Project.

## Competing Interests

The authors declare there are no competing interests.

## Author Contributions

- Hammam Alshazly and Christoph Linse conceived and designed the experiments, performed the experiments, analyzed the data, performed the computation work, prepared figures and/or tables, authored or reviewed drafts of the paper, and approved the final draft.
- Mohamed Abdalla, Erhardt Barth and Thomas Martinetz analyzed the data, authored or reviewed drafts of the paper, and approved the final draft.

## Data Availability

The code for the architectures are available at GitHub: https://github.com/Criscraft/CovidResNet_CovidDenseNet.

The SARS-CoV-2 CT scan dataset, which contains CT images for three different classes, is available at Kaggle: https://www.kaggle.com/plameneduardo/a-covid-multiclass-dataset-of-ct-scans.

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
