# Peer review of "COVID-Nets: deep CNN architectures for detecting COVID-19 using chest CT scans"

_PeerJ Computer Science, doi:10.7717/peerj-cs.655_

## Round 0.1 · original submission · Major Revisions

The paper has a nice contribution. However, some parts need to be detailed with respect to the reviewers' comments.

·

Basic reporting

No comment

Experimental design

No comments

Validity of the findings

No comment

Additional comments

Authors need to add number of cases and death tolls
Authors need to add a paragraph on the increase in cases despite massive vaccination globally
A summary table of related studies showing number of dataset use, where it's obtain (I.e. online or hospital), model use (Resnet, AlexNet, GoogleNet etc) and result interms of sensitivity, AUC, accuracy etc and reference is required.
A flow chart can be added to the methodology section in terms of data collection, processing, data split, performance evaluation etc.
Figure 5, 7, 9, 11 can be move to supplementary file since they are listed in the tables.
Authors need to provide sound and scientific reason why they choose 60-40 for training and validation or testing. Some of the studies vary data partition (60-40, 70-30, 80-20, 90-10)

Reviewer 2 ·

Basic reporting

The English of this manuscript is okay but should be polished.
Please add some methodological details to the abstract section. Which parts are novel?
COVID studies are being studied intensely. The advantages over the current situation should be explained in support of the literature.
Contributions must be rewritten.
Related works section should be supported with a table. Also add please: Coronavirus (covid-19) classification using ct images by machine learning methods, Coronavirus (covid-19) classification using deep features fusion and ranking technique, Classification of Coronavirus (COVID‐19) from X‐ray and CT images using shrunken features, Classification of COVID-19 in Chest CT Images using Convolutional Support Vector Machines.
Also manuscript structure shuld be improved.

Experimental design

The novelty of the methods should be presented more clearly in the methodology section.
Two novel CNN architectures are proposed. However, this novelty is not mentioned in the method section. In this case, the method section is almost nonexistent. Please include a method section that highlights and proves the novelty.

Validity of the findings

More COVID datasets should be used to validate the proposed model.
The proposed method should be compared with the state-of-the-art.

Additional comments

Two novel CNN architectures are proposed. However, this novelty is not mentioned in the method section. In this case, the method section is almost nonexistent. Please include a method section that highlights and proves the novelty.

·

Basic reporting

no comment

Experimental design

no comment

Validity of the findings

no comment

Additional comments

The authors proposed two novel deep CNN architectures (CovidResNet and CovidDenseNet) for automated COVID-19 detection using chest CT scans. The models have been developed and validated for the multi-class and binary classification tasks to differentiate COVID-19 patients from non-COVID-19 viral infections as well as healthy subjects. The SARS-CoV-2 CT-scan dataset has been used to validate the proposed models. Promising performances have been achieved.

The paper is well-structured and easy to read. The proposed models are different from most of the state-of-the-art methods which apply baseline models and being more efficient. However, I have some concerns and questions:

1- Table 3 shows that the dataset is not balanced. How did the proposed method deal with this problem in order to avoid biased experimental results? Is this problem handled in the augmentation steps?
2- The authors reported: “We choose 59.5% of the subjects for training and 40.5% for testing, such that the amount of training images is 60% and 40%, respectively”. In this sentence, did the author mean “training and test images is 60% and 40%, respectively “?
3- Why the authors didn’t apply the 10 fold-cross validation strategy to better evaluate the system instead of the previous split and to compare it with other state-of-the-art methods following the same protocol?
4- In the transfer learning stage, the authors report that “In CovidDenseNet the adapter layers and the last layer are randomly initialized “. Could the authors explain why a random initialization is carried out with this model, contrarily to CovidResNet model?
5- Why do the proposed architectures (CovidDenseNet and CovidResNet) need more epochs to converge (150 epochs vs 100 epochs for the baseline).
6- The authors should provide a comparison between the baseline and the proposed COVID-Nets when dealing with the same number of epochs (100 and 150 epochs) to get a fair comparison and also to give more detail about the convergence area of each model.
7- The authors added two different ensemble combinations for improving the overall diagnostic performance. However, there is no explanation about the type of the applied ensemble model (e.g. majority voting, weighted averaging, or other).
8- The paper presents a comparative study of the proposed models with two baselines (fine-tuned ResNet and DensNet models). Nevertheless, the comparative study needs to be extended to other methods from the literature applying different pre-trained models that achieved very high accuracies using transfer learning strategy (i.e. VGG-16, Inception-ResNetV2).

---

## Round 0.2 · Minor Revisions

One of the reviewer asks for the revision. Please increase the number of images and give a discussion on the confusion matrix. Then, we will finalize the process soon.

·

Basic reporting

Number of dataset is very significant in machine learning and has to be taken into account.

Experimental design

After discussing about the evaluation performance formula. Authors should add a table on confusion matrix

Validity of the findings

The number of images has to be increased

Additional comments

Comments
Authors have addressed all the comments but the article can be upgraded with the comments below

To compare state of art with currents studies, authors need to show the number of images use for each article and make realistic comparison. Number of dataset is very significant in machine learning. A model that train using 1000 images cannot be compare with a model that is train using 5000 images. This is very important.

After discussing about the evaluation performance formula. Authors should add a table on confusion matrix

Reviewer 2 ·

Basic reporting

Authors have solved my issues.

Experimental design

Authors have solved my issues.

Validity of the findings

Authors have solved my issues.

Additional comments

Authors have solved my issues.

·

Basic reporting

no comment

Experimental design

no comment

Validity of the findings

no comment

Additional comments

The authors have addressed all of my concerns.

---

## Round 0.3 · accepted · Accept

The comments have been addressed. We are pleased to inform you that your manuscript has been accepted for publication.

·

Basic reporting

The manuscript is acceptable since all corrections have been made.

Experimental design

The article is acceptable since all corrections have been made.

Validity of the findings

The article is acceptable since all corrections have been made.